# TRACE: Trajectory Recovery for Continuous Mechanism Evolution in Causal Representation Learning

**Shicheng Fan**[1]  **Kun Zhang**[2][3]  **Lu Cheng**[1]

## Abstract

Temporal causal representation learning methods assume that causal mechanisms switch instantaneously between discrete domains, yet real-world systems often exhibit *continuous* mechanism transitions. For example, a vehicle's dynamics evolve gradually through a turning maneuver, and human gait shifts smoothly from walking to running. We formalize this setting by modeling transitional mechanisms as convex combinations of finitely many *atomic mechanisms*, governed by time-varying mixing coefficients. Our theoretical contributions establish that both the latent causal variables and the continuous mixing trajectory are jointly identifiable. We further propose **TRACE**, a Mixture-of-Experts framework where each expert learns one atomic mechanism during training, enabling test-time recovery of mechanism trajectories, including intermediate mechanism states never observed during training. Experiments on synthetic and real-world data demonstrate that TRACE recovers mixing trajectories with up to 0.99 correlation, substantially outperforming discrete-switching baselines.[1]

## 1. Introduction

Causal representation learning (CRL) (Schölkopf et al., 2021; Zhang et al., 2024) seeks to uncover latent causal variables and their relationships from high-dimensional observations. In the context of temporal data, existing CRL approaches (e.g., Yao et al., 2022a; Song et al., 2023; Lippe et al., 2022b; Chen et al., 2024) assume that the states of underlying causal mechanisms are discrete, echoing clas-

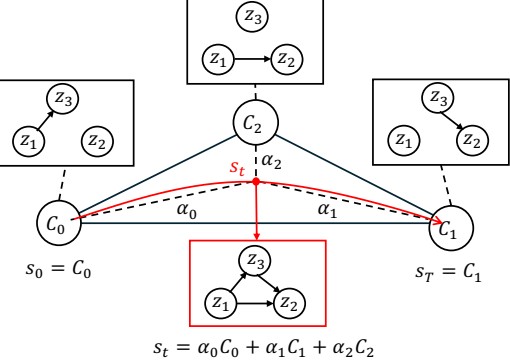

*Figure 1.* Continuous mechanism transitions as trajectories through a simplex. Atomic mechanisms $\{C_0, C_1, C_2\}$ at the vertices each define a causal graph over latent variables $z_1, z_2, z_3$ (boxed; edge colors match the corresponding vertex). Any interior point $s_t = \sum_k \alpha_k C_k$ with $\alpha_k \geq 0$, $\sum_k \alpha_k = 1$ induces a mixed graph (bottom box). The red and dashed curves are distinct trajectories sharing the same endpoints, which discrete-switching models cannot distinguish.

sical assumptions in causal discovery from nonstationary data (Huang et al., 2020). However, in real-world systems, causal mechanisms typically evolve continuously over time. Take vehicle dynamics as an example. A turning maneuver involves gradual shifts in the causal relationships between steering, velocity, and lateral acceleration as the vehicle transitions from straight-line driving through curve entry to steady-state cornering. Similarly, human gait transitions from walking to running (Hreljac, 1993; Diedrich & Warren Jr, 1995) involve continuous reorganization of limb coordination and ground contact dynamics, not instantaneous switches between discrete movement patterns. Understanding these transitions has profound implications. For example, identifying critical points along the mechanism evolution could enable predictive control, early anomaly detection, or targeted intervention before system behavior fundamentally shifts.

When mechanism transitions are continuous, the space of mechanisms often admits a low-dimensional structure. Just as any color can be represented as a mixture of RGB primaries, or a document as a mixture over topics (Blei et al., 2003), intermediate causal mechanisms can be expressed

[1]University of Illinois at Chicago [2]Carnegie Mellon University [3]Mohamed bin Zayed University of Artificial Intelligence. Correspondence to: Shicheng Fan <sfan25@uic.edu>, Lu Cheng <lucheng@uic.edu>.

*Proceedings of the 43rd International Conference on Machine Learning*, Seoul, South Korea. PMLR 306, 2026. Copyright 2026 by the author(s).

[1]Code: https://github.com/shichengf/trace.

as convex combinations of a finite set of *atomic mechanisms*, which are canonical causal graphs that serve as basis elements spanning the mechanism space. This key insight, that a potentially infinite-dimensional mechanism space can be characterized by finitely many atomic mechanisms, motivates a **geometric** view: each mechanism state $s_t$ is a point in a continuous simplex $\mathcal{S}$, atomic mechanisms $\{C_0, \ldots, C_{K-1}\}$ are anchor points (vertices of the simplex) learned from pure-domain data, and the causal graph at any interior point is a convex combination of atomic mechanisms graphs with mixing coefficients $\boldsymbol{\alpha}(t)$. We illustrate this continuous formulation in Figure 1.

The core research question is then: **how can we learn representations that capture the *continuous* evolution of causal mechanisms?** This requires addressing two key challenges. First, we must learn a latent space where mechanism-specific structure is preserved, a challenge connected to identifiability in nonlinear independent component analysis (ICA) (Hyvarinen & Morioka, 2016; 2017; Khemakhem et al., 2020), enabling the encoder to produce representations that reflect which atomic mechanisms are active. Second, given observations from a system undergoing mechanism transition, we must recover the continuous mixing trajectory $\boldsymbol{\alpha}(t)$ through mechanism space, even without ground-truth labels for intermediate states.

**Our Approach.** We introduce **TRACE**, a two-stage framework for recovering continuous mechanism trajectories. Our key insight is that if mechanisms are convex combinations of finitely many atomic mechanisms, then a Mixture-of-Experts architecture (Jacobs et al., 1991; Shazeer et al., 2017) provides a natural implementation: each expert learns the transition dynamics of one atomic mechanism, and their combination captures intermediate mechanisms. In Stage 1, we train a shared encoder alongside domain-specific experts on pure-domain data. In Stage 2, we recover the mixing coefficients $\boldsymbol{\alpha}(t)$ at test time via a least-squares procedure that projects learned representations onto the span of mechanism-specific basis vectors. This formulation enables inference of mechanism trajectories for transitions never observed during training.

**Contributions. (1) Problem.** We formalize CRL with continuously evolving mechanisms, modeling intermediate states as convex combinations of canonical atomic mechanisms. **(2) Method.** We introduce TRACE, a two-stage framework using Mixture-of-Experts (Jacobs et al., 1991; Shazeer et al., 2017; Dai et al., 2024) to learn atomic mechanisms and recover continuous mixing trajectories at test time. **(3) Theory.** We establish identifiability guarantees: latent variables are recoverable up to permutation and component-wise transformation, and mixing trajectories admit finite-sample error bounds that improve with trajectory smoothness. **(4) Experiments.** We validate on synthetic, semi-synthetic, and real-world data, demonstrating accurate

recovery of mechanism trajectories.

## 2. Related Work

**Causal Representation Learning.** CRL aims to recover latent causal variables from high-dimensional observations (Schölkopf et al., 2021; Peters et al., 2017). A central challenge is identifiability: without constraints, disentangled representations cannot be learned unsupervised (Locatello et al., 2019). This motivates leveraging auxiliary information such as temporal structure (Hyvarinen & Morioka, 2016), domain labels (Hyvarinen et al., 2019), or interventions (Khemakhem et al., 2020). In temporal settings, Temporally Disentangled Representation Learning (TDRL) (Yao et al., 2022a) achieves identifiability via time-delayed dependencies, and Nonstationary Causal Temporal Representation Learning (NCTRL) (Song et al., 2023) extends this to nonstationary settings with discrete mechanism switches. Recent work addresses complementary challenges: instantaneous effects (Lippe et al., 2022a; Li et al., 2024), non-invertible mixing (Chen et al., 2024), and sparse transitions (Song et al., 2024). More recent advances explore multi-modal observations (Sun et al., 2025b), sparse mixing for disentanglement (Li et al., 2025a), and latent hierarchical structures (Prashant et al., 2025). However, all these methods assume mechanisms switch discretely between a finite set. Our work addresses a distinct problem: recovering *continuous* trajectories through mechanism space, where intermediate states arise as weighted combinations of atomic mechanisms.

**Mixture-of-Experts.** MoE architectures (Jacobs et al., 1991; Shazeer et al., 2017) combine specialized models via learned gating. While recent advances focus on efficiency and expressivity in language models (Dai et al., 2024; Muennighoff et al., 2025), these methods lack causal structure. MoCE (Gao et al., 2025) introduces causal experts but addresses static deconfounding rather than temporal transitions. Our work repurposes MoE for continuous mechanism transitions: each expert learns one atomic mechanism's dynamics, and the learned representations enable recovery of mixing trajectories via least-squares projection.

**Continuous-Time and Regime-Switching Methods.** Neural ODEs (Chen et al., 2018; Rubanova et al., 2019) model continuous-time latent dynamics but assume fixed dynamical functions. Regime-switching models (Fox et al., 2011; Dong et al., 2020; Xu et al., 2025) allow transitions between discrete regimes but do not parameterize intermediate mechanisms. Neither addresses causal identifiability of mechanism transitions. Assuming intermediate mechanisms admit a convex-combination representation over a finite set of atomic mechanisms, TRACE is, to our knowledge, the first framework to provide joint identifiability guarantees for both the latent causal variables and the continuous mixing trajectory, with finite-sample error bounds.

# 3. Problem Formulation

We formalize the problem of learning continuous mechanism transitions in temporal causal systems. We extend standard temporal CRL by allowing mechanisms to vary continuously rather than discretely between domains.

## 3.1. Generative Model

We consider temporal data generated from a latent dynamical system with time-varying transition dynamics (Figure 1). Let $\mathbf{z}_t \in \mathbb{R}^d$ denote the latent causal variables at time $t$ and $\mathbf{x}_t \in \mathbb{R}^p$ denote the corresponding high-dimensional observation. The data generation process follows:

$$\mathbf{z}_t = f(\mathbf{z}_{t-L:t-1}; \theta_t) + \boldsymbol{\epsilon}_t, \qquad (1)$$
$$\mathbf{x}_t = g(\mathbf{z}_t), \qquad (2)$$

where $f$ is a nonlinear transition function with time-varying parameters $\theta_t$, $L \geq 1$ is the lag order so that the latent process is an $L$-th order Markov process, $g : \mathbb{R}^d \to \mathbb{R}^p$ is an invertible nonlinear mixing function, and $\boldsymbol{\epsilon}_t$ is independent noise.

**Prior Work: Discrete Mechanisms.** Existing temporal CRL methods (Yao et al., 2022a; Song et al., 2023) assume $\theta_t \in \{\theta^{(0)}, \dots, \theta^{(K-1)}\}$ switches discretely among $K$ fixed parameter configurations, with instantaneous transitions between them.

**Our Formulation: Continuous Mechanisms.** We instead model the mechanism parameters as evolving continuously via convex combinations of $K$ *atomic mechanisms*:

$$\theta_t = \sum_{k=0}^{K-1} \alpha_k(t) \cdot \theta^{(k)}, \qquad (3)$$

where $\boldsymbol{\alpha}(t) \in \Delta^{K-1}$ lies in the probability simplex. We adopt simplex constraints for interpretability; unconstrained combinations are also identifiable but less interpretable. We quantify this trade-off empirically in Appendix E.9: on out-of-distribution transitions, the simplex achieves Weight Corr 0.956 versus 0.971 for the unconstrained variant, while eliminating 18% physically invalid (negative) mixing values. This formulation captures gradual transitions that discrete models cannot represent.

## 3.2. Mechanism Parameterization

We instantiate the mechanism parameters $\theta^{(k)}$ as weight matrices $W^{(k)} \in \mathbb{R}^{d \times d}$, with transition function $f(\mathbf{z}_{t-1}; W) = \sigma(W\mathbf{z}_{t-1})$, where $\sigma$ is a component-wise nonlinear activation (e.g., LeakyReLU). The mechanism-specific causal structure is encoded in $W^{(k)}$, which determines causal edge strengths between latent variables.

**Assumption 3.1** (Distinguishable Mechanisms)**.** The transition matrices $\{W^{(k)}\}_{k=0}^{K-1}$ are mutually distinguishable:

no $W^{(k)}$ can be expressed as a convex combination of the others. This is necessary for the "atomic" nature of mechanisms—otherwise, a representable mechanism would be decomposable into more fundamental components, violating atomicity. This imposes the capacity constraint $K \leq d + 1$, since identifiability requires that for each latent dimension $i$, the $K - 1$ row vectors $\{(W^{(k)} - W^{(0)})_{i,:}\}_{k=1}^{K-1}$ are linearly independent in $\mathbb{R}^d$. Under mild regularity conditions, this independence transfers to the basis matrix used in Theorem 4.2 (Appendix A.3). The bound $K \leq d + 1$ is *sufficient*, not necessary: when active mechanisms at test time are sparse or structured, recovery succeeds well beyond this regime, with the binding constraint tightening to $K_{\text{active}} \leq d + 1$ rather than $K_{\text{total}} \leq d + 1$ (Appendix E.11).

Under Eq. (3), the effective transition matrix becomes $W(t) = \sum_{k=0}^{K-1} \alpha_k(t) \cdot W^{(k)}$, yielding:

$$f(\mathbf{z}_{t-1}; W(t)) = \sigma(W(t)\mathbf{z}_{t-1}), \qquad (4)$$

where $\sigma$ is shared across mechanisms while the causal structure $W(t)$ evolves continuously. Each $W^{(k)}$ plays the role of a lagged-causal-effect (transition) matrix in a Dynamic Bayesian Network, so $\boldsymbol{\alpha}(t)$ parameterizes a time-varying DBN with continuously interpolated transition weights (Appendix E.13).

## 3.3. Problem Statement

During training, we observe labeled data from $K$ discrete domains $\{\mathcal{D}_k\}_{k=0}^{K-1}$, where trajectories in $\mathcal{D}_k$ are generated with fixed mechanism $W^{(k)}$ (i.e., $\alpha_k = 1$ and $\alpha_j = 0$ for $j \neq k$). At inference time, we observe unlabeled trajectories $\{\mathbf{x}_t\}_{t=1}^{T}$ undergoing continuous mechanism transitions, where $\boldsymbol{\alpha}(t)$ varies smoothly over time. Neither ground-truth latent states nor mechanism labels are available during inference.

Our goal is twofold: **first**, to learn latent representations $\hat{\mathbf{z}}_t$ that are identifiable up to permutation and component-wise transformation; and **second**, to recover the continuous mixing trajectory $\{\boldsymbol{\alpha}(t)\}_{t=1}^{T}$ from test observations, thereby enabling identification of mechanism transition dynamics and critical transition points.

# 4. Identifiability Theory

We establish two complementary identifiability results. First, latent causal variables are recoverable under conditions inherited from temporal CRL (Theorem 4.1). Second, as our main theoretical contribution, we prove that the *continuous mixing trajectory* $\boldsymbol{\alpha}(t)$ is also recoverable, with finite-sample error bounds that improve with trajectory smoothness (Theorems 4.2–4.3). Complete proofs are in Appendix A, with a summary of all assumptions in Appendix A.10.

## 4.1. Identifiability of Latent Causal Processes

**Theorem 4.1** (Identifiability of Latent Variables). *Suppose* $\mathbf{x}_t = g(\mathbf{z}_t)$ *where* $g$ *is invertible, and the conditional distribution* $p(z_{k,t} \mid \mathbf{z}_{t-1})$ *varies across* $K$ *domains. Under sufficient variability conditions (Appendix A.2), any learned representation* $\hat{\mathbf{z}}_t$ *satisfying conditional independence constraints is related to the true latents by* $\hat{z}_i = h_i(z_{\pi(i)})$, *where* $\pi$ *is a permutation and each* $h_i$ *is strictly monotonic.*

This result builds on the identifiability framework of Yao et al. (2022a), which requires abstract "sufficient variability" conditions on the conditional distributions. We provide a concrete, verifiable criterion: Assumption 3.1 (linear independence of pairwise differences $\{W^{(k)} - W^{(0)}\}_{k=1}^{K-1}$) implies the variability condition under a mild row-wise non-degeneracy requirement, with capacity limit $K \leq d + 1$ (Lemma A.3, Appendix A.3).

## 4.2. Recovery of Mixing Coefficients

While Theorem 4.1 guarantees recovery of latent variables, it does not address how to infer the mixing trajectory $\boldsymbol{\alpha}(t)$ from these representations. We now establish recovery guarantees for this previously unstudied problem.

Let $\hat{\boldsymbol{\mu}}^{(k)}$ denote the conditional expectation under domain $k$, $\delta\hat{\boldsymbol{\mu}}^{(k)} = \hat{\boldsymbol{\mu}}^{(k)} - \hat{\boldsymbol{\mu}}^{(0)}$ the shift relative to baseline, and $\hat{B} = [\delta\hat{\boldsymbol{\mu}}^{(1)}, \ldots, \delta\hat{\boldsymbol{\mu}}^{(K-1)}]$ the basis matrix with minimum singular value $\sigma_{\min} > 0$.

**Theorem 4.2** (Pointwise Recovery). *Under Theorem 4.1's conditions, the least-squares estimator* $\hat{\boldsymbol{\alpha}}(t) = \hat{B}^{\dagger}(\hat{\mathbf{z}}_t - \hat{\boldsymbol{\mu}}^{(0)})$ *satisfies*

$$\|\hat{\boldsymbol{\alpha}}(t) - \boldsymbol{\alpha}(t)\| \leq \frac{1}{\sigma_{\min}} \left(\|\hat{\boldsymbol{\epsilon}}_t\| + \delta_{\text{approx}}\right), \quad (5)$$

*where* $\hat{\boldsymbol{\epsilon}}_t$ *is observation noise and* $\delta_{\text{approx}} = O(\epsilon^2)$ *with* $\epsilon := \max_k \|W^{(k)} - W^{(0)}\|$ *bounding the perturbation magnitude.*

The pointwise bound treats each time step independently and does not improve with trajectory length $T$. By exploiting temporal smoothness, we obtain stronger guarantees.

**Theorem 4.3** (Smooth Trajectory Recovery). *Consider a test trajectory of length* $T$. *Suppose additionally that the trajectory has bounded total variation* $\text{TV}(\boldsymbol{\alpha}^*) \leq V$ *and noise is i.i.d. sub-Gaussian with parameter* $\sigma$. *Consider the regularized estimator:*

$$\hat{\boldsymbol{\alpha}}^{\text{smooth}} = \arg\min_{\boldsymbol{\alpha}_{1:T}} \mathcal{L}_{\text{data}} + \lambda\mathcal{L}_{\text{smooth}}, \quad (6)$$

*where* $\mathcal{L}_{\text{data}} = \sum_{t=1}^{T} \|\hat{\mathbf{z}}_t - \hat{\boldsymbol{\mu}}^{(0)} - \hat{B}\boldsymbol{\alpha}_t\|^2$ *and* $\mathcal{L}_{\text{smooth}} = \sum_{t=1}^{T-1} \|\boldsymbol{\alpha}_{t+1} - \boldsymbol{\alpha}_t\|^2$. *With optimally chosen* $\lambda \asymp T^{1/3}$,

*the mean squared error satisfies*

$$\frac{1}{T} \sum_{t=1}^{T} \mathbb{E}\|\hat{\boldsymbol{\alpha}}_t^{\text{smooth}} - \boldsymbol{\alpha}_t^*\|^2$$
$$= O\left(\left(\frac{V}{T}\right)^{2/3} \frac{\sigma^{2/3}(K-1)^{1/3}}{\sigma_{\min}^{2/3}} + \frac{\delta_{\text{approx}}^2}{\sigma_{\min}^2}\right). \quad (7)$$

*The ratio* $V/T$ *represents the average per-step variation; smaller values (slower transitions or denser sampling) yield tighter bounds.*

The $O(T^{-2/3})$ rate is minimax optimal for bounded total variation signals; smoother transitions (smaller $V$) admit better recovery.

*Remark* 4.4 (Scale Calibration). Recovered coefficients may exhibit systematic scale distortion, correctable via boundary conditions when available. Qualitative structure (monotonicity, transition timing) is preserved without calibration; see Appendix A.4. We validate this empirically in **Section** 6.2.

*Remark* 4.5 (Verifiability). All quantities in the error bounds are empirically measurable: $\sigma_{\min}$ from the basis matrix, $\|\hat{\boldsymbol{\epsilon}}_t\|$ and $\sigma$ from validation residuals, $V$ from recovered trajectories, and $\delta_{\text{approx}}$ from mixed-domain data when available. This verifiability distinguishes our analysis from purely asymptotic results. We demonstrate this in **Section** 6.2.

*Remark* 4.6 (Geometric Bottleneck). The error bound's dependence on $\sigma_{\min}$ reveals a geometric bottleneck: as the number of *active* mechanisms $K_{\text{active}}$ approaches $d + 1$, the corresponding basis columns become increasingly collinear, degrading $\boldsymbol{\alpha}(t)$ recovery even when the model correctly learns each atomic mechanism. This bottleneck affects inference-time decomposition rather than learning quality—the transition structure $W(t)$ remains recoverable, but projecting onto near-collinear bases becomes ill-conditioned. We empirically validate this distinction in **Section** 6.5, and show in Appendix E.11 that the $K \leq d + 1$ condition is sufficient but not necessary, so $K_{\text{total}}$ may safely exceed $d + 1$ when transitions involve few simultaneously active mechanisms.

## 5. Method

We present TRACE, a two-stage framework (Figure 2) for learning continuous mechanism transitions. Stage 1 trains a shared encoder and domain-specific experts to solve the CRL problem on pure-domain data. Stage 2 recovers mechanism trajectories at test time by solving for mixing coefficients via a principled least-squares procedure.

### 5.1. Stage 1: MoE-based Representation Learning

The first stage learns disentangled latent representations from labeled domain data $\{\mathcal{D}_k\}_{k=0}^{K-1}$, where each domain $k$

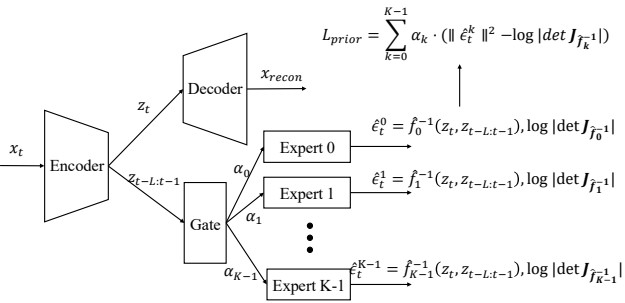

*Figure 2.* Architecture of TRACE. **Stage 1 (Training):** Shared encoder and one-hot gating route to domain-specific experts. **Stage 2 (Inference):** Mixing coefficients $\boldsymbol{\alpha}$ recovered via least-squares projection (Algorithm 1).

corresponds to a distinct atomic mechanism.

Our central hypothesis, that any mechanism state can be expressed as a convex combination of finitely many atomic mechanisms, naturally suggests a Mixture-of-Experts (Jacobs et al., 1991; Shazeer et al., 2017) architecture. Each expert learns to model one atomic mechanism, and the learned representations enable recovery of mixing coefficients $\boldsymbol{\alpha}(t)$ at inference time. This design ensures a one-to-one correspondence between learned experts and identifiable atomic mechanisms, enabling principled trajectory recovery in Stage 2.

We instantiate this idea by extending the sequential VAE framework (Kingma & Welling, 2014; Jimenez Rezende et al., 2014) with an MoE structure where each expert specializes in one domain's causal transition dynamics.

**Shared Encoder.** A shared encoder $g_\phi^{-1}$ maps observations to latent variables across all domains, enforcing the assumption that while mechanisms vary, the underlying causal variables $\mathbf{z}_t$ remain consistent; only their relationships change.

Given observation $\mathbf{x}_t$ and its temporal context $\mathbf{x}_{t-L:t-1}$, the encoder outputs Gaussian posterior parameters:

$$\boldsymbol{\mu}_t, \boldsymbol{\sigma}_t = g_\phi^{-1}(\mathbf{x}_t, \mathbf{x}_{t-L:t-1}), \qquad (8)$$

with latent samples obtained via reparameterization: $\mathbf{z}_t = \boldsymbol{\mu}_t + \boldsymbol{\sigma}_t \odot \boldsymbol{\epsilon}$, where $\boldsymbol{\epsilon} \sim \mathcal{N}(\mathbf{0}, \mathbf{I})$.

**Domain-Specific Experts.** Each domain $k$ is associated with an expert network modeling the causal transition dynamics. Following the normalizing flow formulation (Dinh et al., 2017; Papamakarios et al., 2017; 2021), each expert learns an inverse transition function $\hat{f}_k^{-1}$ mapping latent variables to independent noise:

$$\hat{\boldsymbol{\epsilon}}_t = \hat{f}_k^{-1}(\mathbf{z}_t, \mathbf{z}_{t-L:t-1}), \qquad (9)$$

with transition log-density computed via change of variables. During training, deterministic one-hot gating routes each sample to its corresponding expert, ensuring each expert specializes in its domain's dynamics.

**Training Objective.** Let $q(\mathbf{z}_t|\mathbf{x}_t) = \mathcal{N}(\boldsymbol{\mu}_t, \text{diag}(\boldsymbol{\sigma}_t^2))$ denote the encoder posterior. For training data from domain $k$,

we define the objective function as follows:

$$\mathcal{L}_k = \mathbb{E}_{q(\mathbf{z}_t|\mathbf{x}_t)}\big[ \log p(\mathbf{x}_t|\mathbf{z}_t) + \log p_k(\mathbf{z}_t|\mathbf{z}_{t-L:t-1}) - \log q(\mathbf{z}_t|\mathbf{x}_t) \big], \qquad (10)$$

where $p_k(\mathbf{z}_t|\mathbf{z}_{t-L:t-1})$ is the transition prior modeled by expert $k$. Under the identifiability conditions of Theorem 4.1, the learned representation satisfies

$$\hat{z}_i = h_i(z_{\pi(i)}), \qquad (11)$$

where $\pi$ is a permutation and each $h_i$ is a component-wise invertible function.

### 5.2. Stage 2: Mechanism Trajectory Inference

Given identifiable latent representations, we recover mixing coefficients $\boldsymbol{\alpha}(t)$ via the procedure justified by Theorem 4.2. **Constructing the Differential Basis.** For each domain $k$, we compute the conditional expectation $\hat{\boldsymbol{\mu}}^{(k)} = \mathbb{E}[\mathbf{z}_t \mid \mathbf{z}_{t-L:t-1}, \text{domain } k]$ using validation data from that domain. Using domain $k = 0$ as baseline, we compute $\delta\hat{\boldsymbol{\mu}}^{(k)} = \hat{\boldsymbol{\mu}}^{(k)} - \hat{\boldsymbol{\mu}}^{(0)}$ and form the basis matrix $\hat{B} = [\delta\hat{\boldsymbol{\mu}}^{(1)}, \ldots, \delta\hat{\boldsymbol{\mu}}^{(K-1)}]$.

**Linear Least-Squares Solver.** Given a test observation, we encode it using the posterior mean $\mathbf{z}_t = \boldsymbol{\mu}_t$ from the encoder, then recover mixing coefficients by projecting onto the basis matrix:

$$\hat{\boldsymbol{\alpha}}(t) = \text{Proj}_{\Delta^{K-1}}\left(\hat{B}^\dagger(\mathbf{z}_t - \hat{\boldsymbol{\mu}}^{(0)})\right), \qquad (12)$$

where $\hat{B}^\dagger = (\hat{B}^\top \hat{B})^{-1}\hat{B}^\top$ is the pseudoinverse and $\text{Proj}_{\Delta^{K-1}}$ projects onto the probability simplex. By Theorem 4.2, the recovery error is controlled by $1/\sigma_{\min}(\hat{B})$.

To exploit trajectory smoothness, we apply temporal averaging with window size $w$:

$$\bar{\boldsymbol{\alpha}}(t) = \frac{1}{2w+1}\sum_{s=t-w}^{t+w}\hat{\boldsymbol{\alpha}}(s), \qquad (13)$$

which yields the improved $O(T^{-2/3})$ rate of Theorem 4.3. The complete procedure is summarized in Algorithm 1 (Appendix B).

**Assumption Verification.** We validate the convex interpolation assumption via a Kolmogorov-Smirnov test comparing residuals on pure-domain versus transition data (Appendix E.3).

**OOD Generalization.** The least-squares projection naturally generalizes to unseen mechanism combinations. If distribution shift occurs between training and test data, an optional affine adapter can align the distributions; we found this effective empirically though unnecessary for our main experiments (Appendix E.5).

# 6. Experiments

We evaluate TRACE on synthetic dynamical systems, semi-synthetic control environments, and real-world vehicle motion data. Our experiments validate both latent identifiability and continuous mechanism trajectory recovery.

## 6.1. Experimental Setup

**Evaluation Metrics.** Following prior work (Yao et al., 2022a; Song et al., 2023; 2024; Li et al., 2025b), we evaluate latent identifiability via *Mean Correlation Coefficient (MCC)* (Hyvarinen & Morioka, 2016), measuring recovery up to permutation and component-wise transformation. For mechanism recovery, we report *Weight Correlation* (abbreviated *Corr.*), the Pearson correlation between true and inferred mixing trajectories $\alpha(t)$. We use temporal smoothing with window size $w = 5$ throughout. A parameter analysis of $w$ can be found in Appendix E.4.

**Synthetic Data.** We construct a latent dynamical system with $d = 8$ dimensions and $K_{\text{total}} = 5$ atomic mechanisms. Each atomic mechanism $k$ is associated with a sparse perturbation $\delta W^{(k)}$ modifying a single edge in the causal transition graph, ensuring linear independence per Assumption 3.1. Observations are generated via an invertible nonlinear mixing function. Training uses 40,000 trajectories per domain from pure mechanisms. For evaluation, we generate transition trajectories involving $K_{\text{active}} = 3$ atomic mechanisms (e.g., domains $0 \to 2 \to 4$), testing the model's ability to recover mixing coefficients when a subset of trained experts are active. Full details of the data generation process are in Appendix D.1; complete model architectures, hyperparameters, and training schedules are in Appendix D.5.

**Baselines.** We compare against temporal CRL methods: TDRL (Yao et al., 2022a), NCTRL (Song et al., 2023) (hard and soft gating), LEAP (Yao et al., 2022b), iVAE (Khemakhem et al., 2020), and PCL (Hyvarinen & Morioka, 2017). All baselines assume discrete mechanisms. Under our setting (no instantaneous effects, dense transitions, invertible mixing), TDRL and NCTRL represent the state-of-the-art. More recent methods, including IDOL (Li et al., 2024), CaRiNG (Chen et al., 2024), CtrlNS (Song et al., 2024), and CHiLD (Li et al., 2025b), address orthogonal challenges and are not directly comparable; see Appendix C for detailed discussion.

## 6.2. Synthetic Experiments

We evaluate: (1) whether TRACE accurately recovers continuous mixing trajectories, (2) whether scale calibration enables precise quantitative recovery, and (3) whether the theoretical error bounds hold empirically.

**Results.** Table 1 summarizes performance. TRACE achieves the highest weight correlation ($0.94 \pm 0.05$), substantially outperforming NCTRL variants (hard: 0.67, soft:

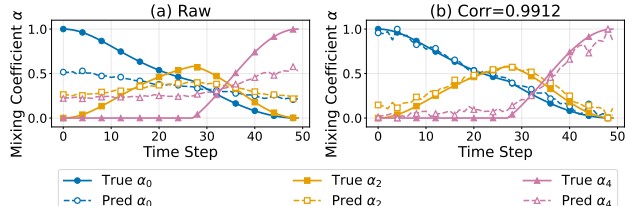

*Figure 3.* Mechanism trajectory recovery with $K_{\text{active}} = 3$ (domains 0, 2, 4). Calibrated predictions accurately track sequential domain activations including the intermediate peak (Corr = 0.99).

*Table 1.* Evaluation on synthetic data. Latent MCC is evaluated on pure-domain test data where all methods operate under their designed conditions; Weight Correlation is evaluated on continuous transition trajectories. "NA" indicates that the method requires discrete domain labels and cannot produce weight estimates during transitions. NCTRL(hard) and NCTRL(soft) use argmax and soft gating strategies, respectively.

| Method | Weight Corr. ↑ | Latent MCC ↑ |
|---|---|---|
| NCTRL(hard) | $0.67 \pm 0.03$ | $0.90 \pm 0.09$ |
| NCTRL(soft) | $0.72 \pm 0.01$ | $0.90 \pm 0.09$ |
| TDRL | NA | $0.89 \pm 0.03$ |
| LEAP | NA | $0.66 \pm 0.05$ |
| iVAE | NA | $0.53 \pm 0.02$ |
| PCL | NA | $0.59 \pm 0.04$ |
| Ours | $\mathbf{0.94 \pm 0.05}$ | $\mathbf{0.96 \pm 0.03}$ |

0.72). Other baselines (TDRL, LEAP, iVAE, PCL) require discrete domain labels and cannot produce trajectory estimates during continuous transitions. For latent identifiability, TRACE achieves MCC of 0.96, exceeding all baselines.

**Scale Calibration.** As predicted by Remark 4.4, raw predictions exhibit systematic scale distortion while preserving trajectory shape. Figure 3 demonstrates recovery on a transition with $K_{\text{active}} = 3$ (domains 0, 2, 4). The calibrated trajectory accurately captures both the sequential activation pattern and intermediate peak timing (Corr. = 0.99). Details on two-point calibration are in Appendix E.2.

**Validation of Theoretical Bounds.** We validate Theorem 4.2 by varying noise level $\sigma_\epsilon$ and perturbation magnitude $\|W^{(k)} - W^{(0)}\|_F$ (Table 2). To quantify recovery difficulty, we define the effective signal-to-noise ratio:

$$\text{SNR}_{\text{eff}} := \frac{\sigma_{\min}}{\bar{\epsilon} + \delta_{\text{approx}}}, \qquad (14)$$

where $\bar{\epsilon}$ is the mean latent residual norm and $\delta_{\text{approx}}$ is the first-order approximation error (Theorem 4.2).

Table 2 reports MCC, Corr., and mean absolute error (MAE) between recovered and true mixing coefficients, measured before ($\text{MAE}_{\text{raw}}$) and after ($\text{MAE}_{\text{cal}}$) scale calibration. Our key observations are: (1) Corr. remains high ($> 0.97$) even when MCC degrades, confirming trajectory shape is preserved; (2) calibration reduces MAE by 3–10×; (3) both metrics improve with larger perturbation magnitude, as pre-

*Table 2.* Validation of theoretical predictions. MCC: latent identifiability; Corr: weight trajectory correlation; MAE: mean absolute error; $\text{SNR}_{\text{eff}}$: effective signal-to-noise ratio (Eq. 14).

| **Scheme 1: Vary Noise** (fixed $\|W^{(k)} - W^{(0)}\|_F = 0.5$) | | | | | |
|---|---|---|---|---|---|
| $\sigma_\epsilon$ | 0.01 | 0.05 | 0.10 | 0.20 | 0.50 |
| MCC ↑ | **0.996** | 0.994 | 0.984 | 0.849 | 0.703 |
| Corr ↑ | **0.998** | 0.998 | 0.998 | 0.996 | 0.854 |
| $\text{MAE}_{\text{raw}}$ ↓ | **0.077** | 0.106 | 0.254 | 0.278 | 0.321 |
| $\text{MAE}_{\text{cal}}$ ↓ | **0.028** | 0.026 | 0.030 | 0.026 | 0.127 |
| $\text{SNR}_{\text{eff}}$ ↑ | **0.411** | 0.388 | 0.251 | 0.101 | 0.060 |
| **Scheme 2: Vary Perturbation** (fixed $\sigma_\epsilon = 0.1$) | | | | | |
| $\|W^{(k)} - W^{(0)}\|_F$ | 0.1 | 0.2 | 0.3 | 0.5 | 0.7 |
| MCC ↑ | 0.934 | 0.953 | 0.970 | 0.984 | **0.990** |
| Corr ↑ | 0.972 | 0.996 | 0.997 | 0.998 | **0.998** |
| $\text{MAE}_{\text{raw}}$ ↓ | 0.319 | 0.292 | 0.266 | 0.254 | **0.194** |
| $\text{MAE}_{\text{cal}}$ ↓ | 0.063 | 0.025 | 0.030 | 0.030 | **0.024** |
| $\text{SNR}_{\text{eff}}$ ↑ | 0.043 | 0.047 | 0.093 | 0.251 | **0.312** |

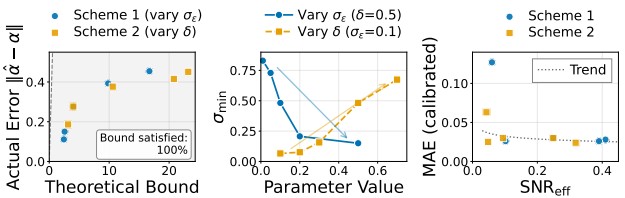

*Figure 4.* Empirical validation of Theorem 4.2. **Left:** Recovery errors vs. theoretical bounds (all points below $y = x$). **Middle:** $\sigma_{\min}$ under varying noise and perturbation. **Right:** MAE vs. $\text{SNR}_{\text{eff}}$.

dicted by the $\sigma_{\min}$ dependence in our bounds.

Figure 4 provides visual confirmation: (Left) all errors fall below theoretical bounds; (Middle) $\sigma_{\min}$ increases with perturbation and decreases with noise; (Right) MAE correlates with $\text{SNR}_{\text{eff}}^{-1}$. Extended analysis is in Appendix E.1.

These results suggest that TRACE accurately recovers continuous mixing trajectories, scale calibration enables precise quantitative recovery, and the theoretical error bounds hold empirically.

## 6.3. Semi-Synthetic Evaluation: Modified CartPole

We evaluate whether TRACE recovers latent variables from high-dimensional pixel observations. We construct a modified CartPole environment with $128 \times 128$ grayscale images, where the latent state $\mathbf{z}_t = (x, v, \theta, \omega)^\top$ represents cart position, velocity, pole angle, and angular velocity. Following TDRL (Yao et al., 2022a), we report MCC on the visually observable components $(x, \theta)$.

**Setup.** We define five domains ($K_{\text{total}} = 5$) where each $W^{(k)} - W^{(0)}$ modifies a single causal edge, with constant Gaussian noise across domains. Under this constant-variance setting, NCTRL's identifiability conditions fail ($\partial^2 \eta / \partial z_{k,t}^2$ is constant across domains), whereas TRACE's conditions hold via linear independence of $\{W^{(k)} - W^{(0)}\}$.

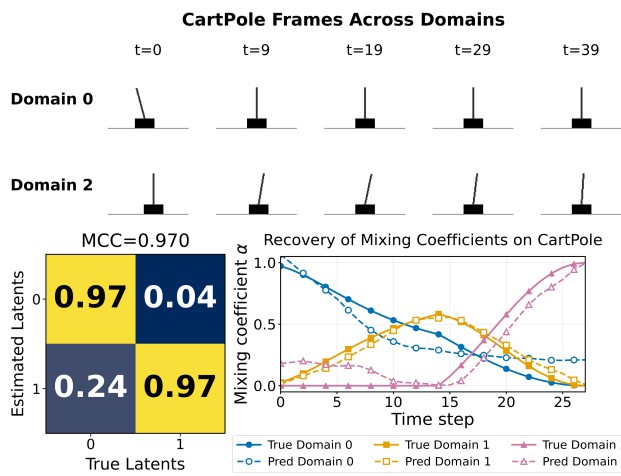

*Figure 5.* Evaluation on modified CartPole. **Top:** Frames across domains showing distinct dynamics. **Bottom left:** Correlation matrix confirms latent identifiability (MCC = 0.970). **Bottom right:** Recovered mixing coefficients (Corr. 0.95) capture the sequential activation pattern.

We train on 1,000 trajectories per domain; full details are in Appendix D.2.

**Results.** Figure 5 shows frames across domains (top), the correlation matrix confirming latent identifiability with MCC = 0.970 (bottom left), and trajectory recovery with Corr. = 0.95 for $K_{\text{active}} = 3$ (bottom right). These results confirm our framework extends to image observations; baseline comparisons follow in Sections 6.4.1–6.4.2.

## 6.4. Real-World Applications

We evaluate TRACE on two real-world datasets: vehicle turning maneuvers (UAVDT (Du et al., 2018)) and human gait transitions (CMU Motion Capture Database[2]). These experiments test whether our framework recovers meaningful mechanism trajectories from natural data.

**Baseline and Evaluation.** Since ground-truth mechanism labels are unavailable, we construct physically-motivated proxies: for the vehicle dataset, the velocity-direction ratio $|v_y|/(|v_x| + |v_y|)$ that is approximately linear in $\alpha_{\text{turn}}$ within the operating range; for the gait dataset, hip joint speed, which increases approximately linearly with the walk-to-run mixing coefficient. These can be viewed as projections of the true mixing coefficients; since we evaluate via Pearson correlation (invariant to monotonic transformations), the resulting error is acceptable (Appendix D.3 and D.4). Among the baselines in Section 6.1, only NCTRL can produce domain assignments without labels; other methods either require labels during transitions or violate their assumptions under our setting.

---

[2]http://mocap.cs.cmu.edu/

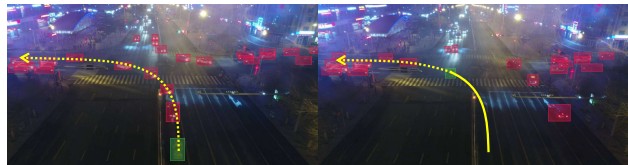

(a) UAV frames with trajectory overlay (yellow curve)

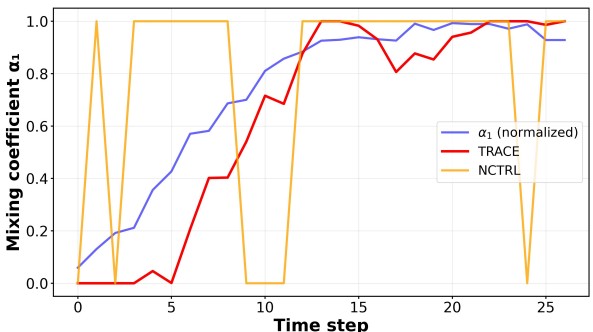

(b) TRACE (Corr. 0.960) vs. NCTRL (Corr. 0.239)

*Figure 6.* Vehicle turning on UAVDT. (a) Sequential frames show the vehicle executing a left turn; the yellow curve indicates the trajectory. (b) TRACE produces smooth $\alpha_1$ trajectories while NCTRL oscillates between discrete states.

### 6.4.1. VEHICLE TURNING (UAVDT)

**Setup.** We extract trajectories during turning maneuvers at major intersections. We define two atomic mechanisms: *horizontal movement* (domain 0) and *vertical movement* (domain 1), training on straight-moving vehicles in each direction. The latent state includes position and velocity ($d = 4$, lag $L = 2$).

**Results.** Figure 6(a) shows UAV frames with the tracked vehicle (green box) and its trajectory (yellow curve) through a left turn. Figure 6(b) compares recovery on this 26-frame sequence: TRACE achieves Corr. of 0.960 with smooth trajectories where $\alpha_1$ gradually increases from 0 to 1, while NCTRL achieves only 0.239, oscillating erratically between discrete states. For reference, simple physics-motivated heuristics on the same trajectories yield substantially worse recovery (centroid tracking 0.911, optical flow 0.902); see Appendix E.12.

### 6.4.2. HUMAN GAIT TRANSITION (CMU MOCAP)

**Setup.** We select walk-to-run transition trials totaling over 1,000 frames. We define two atomic mechanisms: *walking* and *running* ($\alpha_{\mathrm{run}}$), training on pure gait sequences.

**Results.** Figure 7(a) shows skeleton overlays at five time points during a walk-to-run transition, with color gradient (blue→red) indicating temporal progression. Figure 7(b) shows the corresponding trajectory recovery: TRACE achieves Corr. of $0.856 \pm 0.043$ across test trials (displayed:

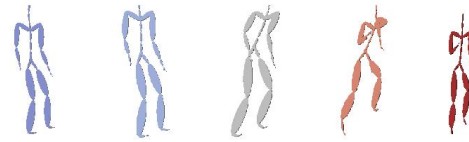

(a) Skeleton snapshots from walking (blue) to running (red)

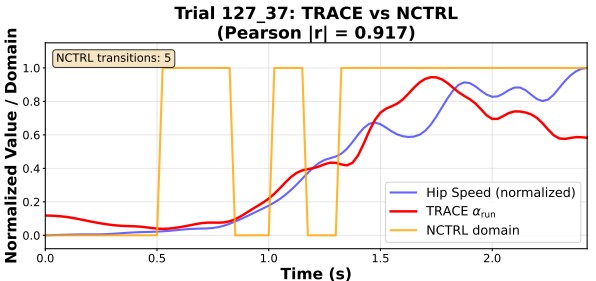

(b) TRACE (Corr. 0.917) vs. NCTRL (Corr. 0.619)

*Figure 7.* Gait transition on CMU MoCap. (a) Skeleton snapshots at five time points; color corresponds to $\alpha_{\mathrm{run}}$ in (b). (b) TRACE tracks proxy smoothly while NCTRL produces 5 spurious discrete transitions.

*Table 3.* Recovery correlation ($\uparrow$). $\mathbf{W}$ remains stable while $\boldsymbol{\alpha}$ degrades with $K_{\mathrm{active}}$.

| $K_{\mathrm{active}}$ | $\boldsymbol{\alpha}$ recovery | | | $\mathbf{W}$ recovery | | |
|---|---|---|---|---|---|---|
| | Simp. | Med. | Comp. | Simp. | Med. | Comp. |
| 2 | .993 | .997 | .979 | 1.000 | 1.000 | 1.000 |
| 3 | .989 | .989 | .971 | 1.000 | 1.000 | 1.000 |
| 4 | .988 | .982 | .919 | .999 | .999 | .999 |
| 5 | .979 | .965 | .835 | .999 | .999 | .999 |
| 6 | .968 | .957 | .692 | .998 | .999 | .999 |
| 7 | .776 | .735 | .459 | .995 | .996 | .998 |

0.917), with smooth $\alpha_{\mathrm{run}}$ tracking the hip speed proxy. NC-TRL produces 5 spurious transitions during this single gait change, unable to represent continuous evolution.

### 6.5. Ablation Study

We disentangle two failure modes: (1) incorrect learning of atomic mechanisms $\{W^{(k)}\}$, and (2) geometric limitations in recovering $\boldsymbol{\alpha}(t)$ via least-squares projection. Full results for $K_{\mathrm{active}} \in \{2, \dots, 10\}$ are in Table 12 (Appendix E.4).

**Setup.** We train with $K_{\mathrm{total}} = 10$ mechanisms and evaluate on $K_{\mathrm{active}} \in \{2, \dots, 7\}$ across three complexity levels (*simple*: sequential, *medium*: overlapping, *complex*: oscillating) with $d = 8$. We report correlation for both $\boldsymbol{\alpha}$ recovery and $W(t) = \sum_k \alpha_k(t) W^{(k)}$ recovery.

**Results.** Table 3 reveals a critical distinction: $W(t)$ recovery remains robust ($> 0.995$) across all conditions, even at $K_{\mathrm{active}} = 7$ where $\boldsymbol{\alpha}$ drops to 0.459, confirming correct learning of mechanism structure. The gap between

*Table 4.* Sample efficiency. Trajectory recovery (Weight Corr) degrades far more gracefully than per-step latent identifiability (MCC), because temporal smoothing (Theorem 4.3) averages out per-step noise.

| Fraction | MCC ↑ | W. Corr ($K=3$) ↑ | W. Corr ($K=5$) ↑ |
|---|---|---|---|
| 100% | 0.963 | 0.986 | 0.979 |
| 20% | $0.886 \pm 0.088$ | $0.968 \pm 0.014$ | $0.955 \pm 0.022$ |
| 10% | $0.752 \pm 0.163$ | $0.960 \pm 0.013$ | $0.941 \pm 0.014$ |
| 1% | $0.622 \pm 0.069$ | $0.688 \pm 0.377$ | $0.640 \pm 0.169$ |

*Table 5.* Recovery beyond the geometric capacity bound ($K_{\text{active}} = 5$, $K_{\text{total}} = 50$). Even at $K/(d+1) = 5.56$, TRACE recovers mechanism trajectories with high correlation.

| $d$ | $K$ | $K/(d+1)$ | $K_{\text{active}}$ | Weight Corr ↑ |
|---|---|---|---|---|
| 8 | 50 | 5.56 | 5 | $0.925 \pm 0.052$ |
| 16 | 50 | 2.94 | 5 | $0.962 \pm 0.010$ |
| 128 | 50 | 0.39 | 5 | $\mathbf{0.989 \pm 0.002}$ |

$W(t)$ and Corr. $\alpha$ widens from $+0.007$ to $+0.538$ as $K_{\text{active}} \to d+1$, validating the geometric bottleneck predicted by Remark 4.6: projection onto near-collinear bases becomes ill-conditioned, affecting inference rather than learning. Our real-world experiments ($K = 2$) in Section 6.4 operate safely within the reliable regime where both metrics exceed 0.95.

### 6.6. Sample Efficiency

We further probe how TRACE behaves when training data is scarce, reducing data from $100\%$ (40,000 trajectories per domain) down to $1\%$ (400 per domain) on the synthetic benchmark.

**Results.** Table 4 reveals an asymmetric robustness pattern: trajectory recovery degrades far more gracefully than per-step latent identifiability. At $10\%$ training data, MCC drops sharply to 0.752, yet Weight Corr at $K_{\text{active}} = 3$ remains 0.960. The per-step signal-to-noise ratio is low ($\sigma_{\min}/\|\hat{\epsilon}_t\| \approx 0.03$), but the temporal smoothing of Theorem 4.3 averages out the noise at the trajectory level. Even at $20\%$ data, Weight Corr exceeds 0.95 across all tested $K_{\text{active}}$ levels, demonstrating practical applicability beyond the large-data regime. The full data scarcity table (with N/domain counts) is in Appendix E.10.

### 6.7. Scaling Beyond the Geometric Capacity Bound

The condition $K \leq d+1$ in Assumption 3.1 is a *sufficient*, not necessary, requirement for trajectory recovery. We empirically stress-test this by training with $K_{\text{total}} = 50$ atomic mechanisms across three latent dimensions.

**Results.** Table 5 shows that even when $K/(d+1) = 5.56$ (50 mechanisms in $d = 8$ dimensions), TRACE achieves

Weight Corr 0.925 for $K_{\text{active}} = 5$. Performance further improves as $d$ grows and the constraint relaxes, reaching 0.989 at $d = 128$. When mechanism perturbations are structurally sparse (each $W^{(k)}$ modifies only two causal edges), Weight Corr at fixed $d = 8$, $K = 50$ improves from 0.925 (dense) to **0.979** (sparse), with variance dropping from $\pm 0.052$ to $\pm 0.006$. The binding empirical constraint is therefore $K_{\text{active}} \leq d+1$, not $K_{\text{total}} \leq d+1$: TRACE remains practical when the mechanism dictionary is large but only a few are active at any time. A mathematical explanation, together with a continuous parameterization $W(t) = h(c_t)$ for the $K_{\text{active}} \gg d$ regime, is given in Appendix E.11.

## 7. Conclusion

Real-world causal mechanisms evolve continuously, but existing causal representation learning methods assume discrete switches between regimes. We presented TRACE, a framework that models mechanism transitions as trajectories on a simplex over $K$ atomic mechanisms, with identifiability guarantees for both latent variables and mixing coefficients. Across synthetic and real-world benchmarks, TRACE achieves 3 to $4\times$ higher correlation with ground-truth dynamics than discrete-switching baselines.

**Limitations.** TRACE has three scope conditions. First, training assumes labeled pure-regime data and a known $K$; in practice, the framework degrades gracefully under domain impurity (Appendix E.7) and mild $K$ misspecification (Appendix E.8). Second, real-world evaluation relies on physically motivated proxies rather than ground-truth labels, with $\sigma_{\min}$ (Appendix E.7) acting as a pre-deployment diagnostic. Third, the theoretical bound $K \leq d+1$ is sufficient for identifiability but indicates a finite-capacity regime; structurally sparse perturbations relax it to $K_{\text{active}} \leq d+1$ (Section 6.7), though a fully unrestricted extension remains open. A detailed discussion of each scope condition is in Appendix E.6, with concrete extensions and downstream applications in Appendix E.14.

## Impact Statement

Understanding continuous mechanism transitions could potentially benefit drug discovery by identifying intervention windows before pathological mechanisms dominate, and robotic manipulation by handling transitions between free-space dynamics and contact-rich regimes. More broadly, tracking how causal relationships evolve continuously opens new directions for mechanism-aware machine learning in scientific applications.

We do not foresee immediate negative societal impacts from this work, as it focuses on foundational methodology rather than application-specific deployment.

## Acknowledgments

This work is supported by the National Science Foundation (NSF) Grant #2312862, NSF-Simons SkAI Institute, NSF CAREER #2440542, NSF #2533996, National Institutes of Health (NIH) #R01AG091762, NSF ACCESS Computing Resources, NAIRR, NRP, a Google Research Scholar Award, and Cisco gift grant.

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

# A. Proofs and Theoretical Foundations

This appendix provides complete proofs for the identifiability results stated in Section 4, along with detailed technical discussion.

## A.1. Background: Identifiability in Temporal Causal Processes

Our identifiability analysis builds upon the theoretical framework established for nonlinear ICA with temporal structure (Hyvarinen & Morioka, 2016; Hyvarinen et al., 2019; Khemakhem et al., 2020).

**Definition A.1** (Component-wise Invertible Transformation). A function $h : \mathbb{R}^d \rightarrow \mathbb{R}^d$ is *component-wise invertible* if $h(\mathbf{z}) = (h_1(z_1), \ldots, h_d(z_d))^\top$ where each $h_i : \mathbb{R} \rightarrow \mathbb{R}$ is strictly monotonic.

**Definition A.2** (Identifiable Latent Causal Processes). Let $\mathbf{x}_t = g(\mathbf{z}_t)$ be observations generated from latent temporal processes. The latent variables are *identifiable* if any learned representation $\hat{\mathbf{z}}_t = \hat{g}^{-1}(\mathbf{x}_t)$ satisfying the model constraints must equal $\mathbf{z}_t$ up to permutation $\pi$ and component-wise invertible transformation $h$: $\hat{z}_i = h_i(z_{\pi(i)})$ for all $i \in \{1, \ldots, d\}$.

The permutation ambiguity is inherent to unsupervised learning, and the component-wise transformation reflects that the scale and nonlinear warping of each dimension cannot be determined from observations alone.

## A.2. Proof of Theorem 4.1: Identifiability of Latent Variables

We adapt the identifiability result from TDRL (Yao et al., 2022a) to our multi-domain setting.

*Proof.* Let $\eta_{k,t}^{(u)} = \log p(z_{k,t} \mid \mathbf{z}_{t-1}, u)$ denote the log-density of component $k$ under domain $u$. Define the derivative vectors:

$$\mathbf{v}_{k,t}^{(u)} \triangleq \left( \frac{\partial^2 \eta_{k,t}^{(u)}}{\partial z_{k,t} \partial z_{1,t-1}}, \ldots, \frac{\partial^2 \eta_{k,t}^{(u)}}{\partial z_{k,t} \partial z_{d,t-1}} \right)^\top \in \mathbb{R}^d, \tag{15}$$

$$\dot{\mathbf{v}}_{k,t}^{(u)} \triangleq \left( \frac{\partial^3 \eta_{k,t}^{(u)}}{\partial z_{k,t}^2 \partial z_{1,t-1}}, \ldots, \frac{\partial^3 \eta_{k,t}^{(u)}}{\partial z_{k,t}^2 \partial z_{d,t-1}} \right)^\top \in \mathbb{R}^d. \tag{16}$$

These vectors capture how the conditional distribution of $z_{k,t}$ depends on the lagged variables $\mathbf{z}_{t-1}$. For the multi-domain setting, we concatenate information across domains:

$$\mathbf{s}_{k,t} \triangleq \left( \mathbf{v}_{k,t}^{(0)\top}, \ldots, \mathbf{v}_{k,t}^{(K-1)\top}, \Delta_1, \ldots, \Delta_{K-1} \right)^\top, \tag{17}$$

$$\dot{\mathbf{s}}_{k,t} \triangleq \left( \dot{\mathbf{v}}_{k,t}^{(0)\top}, \ldots, \dot{\mathbf{v}}_{k,t}^{(K-1)\top}, \dot{\Delta}_1, \ldots, \dot{\Delta}_{K-1} \right)^\top, \tag{18}$$

where $\Delta_u = \frac{\partial^2 \eta_{k,t}^{(u)}}{\partial z_{k,t}^2} - \frac{\partial^2 \eta_{k,t}^{(u-1)}}{\partial z_{k,t}^2}$ captures the change in curvature across adjacent domains.

Suppose $\hat{g}^{-1}$ is any function such that $\hat{\mathbf{z}}_t = \hat{g}^{-1}(\mathbf{x}_t)$ has mutually independent components conditional on $\hat{\mathbf{z}}_{t-1}$. By the change of variables formula, if $\hat{\mathbf{z}} = \phi(\mathbf{z})$ for some diffeomorphism $\phi = \hat{g}^{-1} \circ g$, then

$$\log p(\hat{\mathbf{z}}_t \mid \hat{\mathbf{z}}_{t-1}, u) = \log p(\mathbf{z}_t \mid \mathbf{z}_{t-1}, u) - \log |\det J_\phi(\mathbf{z}_t)|, \tag{19}$$

where $J_\phi$ is the Jacobian of $\phi$.

The conditional independence of $\hat{\mathbf{z}}_t$ given $\hat{\mathbf{z}}_{t-1}$ implies that the cross-derivatives of the transformed log-density must vanish:

$$\frac{\partial^2}{\partial \hat{z}_{i,t} \partial \hat{z}_{j,t}} \log p(\hat{\mathbf{z}}_t \mid \hat{\mathbf{z}}_{t-1}, u) = 0, \quad \forall i \neq j. \tag{20}$$

Expanding this constraint using the chain rule and the linear independence of $\{\mathbf{s}_{k,t}, \dot{\mathbf{s}}_{k,t}\}_{k=1}^d$, one can show that $\phi$ must decompose as $\phi = P \circ h$ where $P$ is a permutation matrix and $h$ is component-wise invertible. $\square$

**Discussion of the Linear Independence Condition.** The sufficient variability condition (linear independence of $\{\mathbf{s}_{k,t}, \dot{\mathbf{s}}_{k,t}\}_{k=1}^n$) is the key requirement that enables identification. This condition fails in two known degenerate cases: (1) i.i.d. processes where $z_{k,t}$ is independent of $\mathbf{z}_{t-1}$, yielding $\mathbf{v}_{k,t}^{(u)} = \mathbf{0}$; and (2) Gaussian additive noise with constant variance across all domains, where second derivatives are constant. Processes with heterogeneous noise variances or non-Gaussian additive noise generically satisfy the condition. In our setting, defining $\delta W^{(k)} := W^{(k)} - W^{(0)}$, the domain-specific differences naturally induce variation in the conditional distributions, ensuring the condition holds when Assumption 3.1 is satisfied.

## A.3. From Parameter Independence to Sufficient Variability

We establish that Assumption 3.1 (linear independence of pairwise differences $\{W^{(k)} - W^{(0)}\}_{k=1}^{K-1}$) implies the sufficient variability condition required by Theorem 4.1. This bridges the gap between parameter-space structure and the function-space condition on conditional distributions.

**Notation.** Throughout this section, we define $\delta W^{(k)} := W^{(k)} - W^{(0)}$ for $k \geq 1$, treating domain 0 as baseline.

**Lemma A.3** (Parameter Independence Implies Sufficient Variability). *Consider the generative model in Section 3 with transition dynamics $\mathbf{z}_t = f(\mathbf{z}_{t-1}; W) + \boldsymbol{\epsilon}_t$, where $f(\mathbf{z}; W) = \sigma(W\mathbf{z})$ for a component-wise activation $\sigma$ and $\boldsymbol{\epsilon}_t \sim \mathcal{N}(\mathbf{0}, \Sigma)$ with $\Sigma \succ 0$. Under Assumption 3.1, if additionally:*

(i) *The activation $\sigma$ is twice differentiable with $\sigma'(x) \neq 0$ for all $x$ in the support of the latent process;*

(ii) *The latent process has sufficient support: for each domain $k$, the distribution of $\mathbf{z}_{t-1}$ has full-rank covariance;*

(iii) *(Row-wise non-degeneracy) For each component $k \in \{1, \ldots, d\}$, the row vectors $\{(\delta W^{(u)})_{k,:}\}_{u=1}^{K-1}$ are linearly independent in $\mathbb{R}^d$, which requires $K - 1 \leq d$;*

*then the sufficient variability condition (linear independence of $\{\mathbf{s}_{k,t}, \dot{\mathbf{s}}_{k,t}\}_{k=1}^{d}$ as defined in Appendix A.2) is satisfied.*

*Remark* A.4 (Capacity Constraint). Condition (iii) implies a fundamental identifiability limit: the number of distinguishable mechanisms cannot exceed the latent dimension plus one, i.e., $K \leq d + 1$. This constraint is satisfied generically: if the entries of $\{\delta W^{(u)}\}$ are drawn from any continuous distribution, row-wise non-degeneracy holds with probability one. When the number of active mechanisms $K_{\text{active}}$ approaches or exceeds this capacity, recovery performance degrades gracefully; we validate this prediction empirically in Section 6.5.

*Proof.* Under Gaussian additive noise, the conditional log-density takes the form:

$$\eta_{k,t}^{(u)} = \log p(z_{k,t} \mid \mathbf{z}_{t-1}, u)$$
$$= -\frac{1}{2\sigma_k^2}\left(z_{k,t} - f_k(\mathbf{z}_{t-1}; W^{(u)})\right)^2 + C, \quad (21)$$

where $f_k$ denotes the $k$-th component of $f$ and $C$ is a normalization constant.

**Step 1: Computing the derivative vectors.** The second-order mixed partial derivative is:

$$\frac{\partial^2 \eta_{k,t}^{(u)}}{\partial z_{k,t} \partial z_{j,t-1}} = \frac{1}{\sigma_k^2} \frac{\partial f_k}{\partial z_{j,t-1}}\bigg|_{W^{(u)}}$$
$$= \frac{1}{\sigma_k^2} \sigma'([W^{(u)}\mathbf{z}_{t-1}]_k) \cdot W_{kj}^{(u)}. \quad (22)$$

Thus the derivative vector $\mathbf{v}_{k,t}^{(u)}$ (defined in Appendix A.2) satisfies:

$$\mathbf{v}_{k,t}^{(u)} = \frac{\sigma'([W^{(u)}\mathbf{z}_{t-1}]_k)}{\sigma_k^2} \cdot (W^{(u)})_{k,:}^{\top}, \quad (23)$$

where $(W^{(u)})_{k,:}$ denotes the $k$-th row of $W^{(u)}$.

**Step 2: Cross-domain differences.** Recall $\delta W^{(u)} = W^{(u)} - W^{(0)}$. The difference between domains $u$ and $0$ yields:

$$\mathbf{v}_{k,t}^{(u)} - \mathbf{v}_{k,t}^{(0)} = \frac{1}{\sigma_k^2}\Big[\sigma'([W^{(u)}\mathbf{z}_{t-1}]_k)(W^{(u)})_{k,:}^{\top}$$
$$- \sigma'([W^{(0)}\mathbf{z}_{t-1}]_k)(W^{(0)})_{k,:}^{\top}\Big]. \quad (24)$$

By Taylor expansion around $W^{(0)}$ and using condition (i) that $\sigma' \neq 0$:

$$\mathbf{v}_{k,t}^{(u)} - \mathbf{v}_{k,t}^{(0)} = \frac{\sigma'([W^{(0)}\mathbf{z}_{t-1}]_k)}{\sigma_k^2}(\delta W^{(u)})_{k,:}^{\top} + O(\|\delta W^{(u)}\|^2). \tag{25}$$

**Step 3: Linear independence.** The vectors $\{\mathbf{v}_{k,t}^{(u)} - \mathbf{v}_{k,t}^{(0)}\}_{u=1}^{K-1}$ inherit linear independence from $\{(\delta W^{(u)})_{k,:}\}_{u=1}^{K-1}$ when the scalar prefactor $\sigma'([W_{\text{base}}\mathbf{z}_{t-1}]_k)/\sigma_k^2$ is nonzero (guaranteed by condition (i)).

However, Assumption 3.1 guarantees linear independence of the *matrices* $\{\delta W^{(1)}, \ldots, \delta W^{(K-1)}\}$ in $\mathbb{R}^{d \times d}$, which does not immediately imply that the row vectors $\{(\delta W^{(u)})_{k,:}\}_{u=1}^{K-1}$ are linearly independent for each fixed $k$. We therefore require an additional regularity condition:

(iii) **(Row-wise non-degeneracy)** For each component $k \in \{1, \ldots, d\}$, the row vectors $\{(\delta W^{(u)})_{k,:}\}_{u=1}^{K-1}$ are linearly independent in $\mathbb{R}^d$.

This condition is satisfied generically: if the entries of $\{\delta W^{(u)}\}$ are drawn from any continuous distribution, row-wise non-degeneracy holds with probability one. Moreover, condition (iii) implies a *capacity constraint*: since each row lies in $\mathbb{R}^d$, we require $K - 1 \leq d$, i.e., the number of distinguishable mechanisms cannot exceed the latent dimension plus one.

Under conditions (i)–(iii), combined with condition (ii) ensuring that $\mathbf{z}_{t-1}$ explores the latent space sufficiently, the concatenated vectors $\{\mathbf{s}_{k,t}\}$ across components and domains span a space of dimension at least $d(K - 1)$.

A similar argument applies to the third-order derivatives $\dot{\mathbf{v}}_{k,t}^{(u)}$, using the second derivative $\sigma''$ and condition (i). The joint linear independence of $\{\mathbf{s}_{k,t}, \dot{\mathbf{s}}_{k,t}\}_{k=1}^{d}$ follows when $\sigma''$ is not identically proportional to $\sigma'$, which holds for common activations including LeakyReLU (where $\sigma'' = 0$ a.e. but the discontinuity at zero provides additional variation) and smooth activations like softplus or tanh.

$\square$

*Remark* A.5 (Verification of Conditions). Condition (i) is satisfied by LeakyReLU ($\sigma'(x) = 1$ for $x > 0$, $\sigma'(x) = 0.2$

for $x < 0$), softplus, tanh, and other common activations. Condition (ii) is a mild regularity assumption that holds when the latent dynamics are ergodic or when training data covers diverse initial conditions. In our experiments, both conditions are satisfied by construction.

### A.4. Preservation of Qualitative Structure

We establish conditions under which the recovered mixing trajectory preserves monotonicity and relative ordering, justifying the claims in Remark 4.4.

**Proposition A.6** (Monotonicity Preservation). *Consider a one-dimensional mixing trajectory $\alpha^*(t) : [0, T] \to [0, 1]$ that is strictly monotonic. Let $\hat{\alpha}(t)$ denote the recovered trajectory via the least-squares estimator of Theorem 4.2. Suppose:*

*(i) The approximation residual satisfies $r(\alpha) = \alpha(1-\alpha) \cdot \mathbf{r}_2 + O(\epsilon^3)$ for some fixed vector $\mathbf{r}_2$ with $\|\mathbf{r}_2\| \le C_2\epsilon^2$;*

*(ii) The perturbation-to-distinguishability ratio satisfies $\epsilon^2/\sigma_{\min} < 1/2$.*

*Then, in the noiseless case ($\hat{\boldsymbol{\epsilon}}_t = \mathbf{0}$), the recovered trajectory $\hat{\alpha}(t)$ is strictly monotonic with the same direction as $\alpha^*(t)$.*

*Proof.* Define the recovery map $\Phi : \alpha \mapsto \hat{\alpha} = \alpha + g(\alpha)$, where $g(\alpha) = \hat{B}^\dagger r(\alpha)$.

**Preliminary: From Parameter to Expectation Independence.** Assumption 3.1 establishes linear independence of $\{W^{(k)} - W^{(0)}\}_{k=1}^{K-1}$ in parameter space. We require that this independence transfers to the conditional expectation differences $\{\delta\boldsymbol{\mu}^{(k)}\}$ forming the basis matrix $B$. By Taylor expansion around $W^{(0)}$:

$$\delta\boldsymbol{\mu}^{(k)} = \mathbb{E}\left[\text{diag}(\sigma'(W^{(0)}\mathbf{z}_{t-1})) \cdot (W^{(k)} - W^{(0)})\mathbf{z}_{t-1}\right] + O(\|W^{(k)} - W^{(0)}\|^2). \quad (26)$$

Linear independence of $\{\delta\boldsymbol{\mu}^{(k)}\}$ follows when $\sigma' \ne 0$ (condition i) and the latent covariance is non-degenerate (condition ii), per Lemma A.3.

**Step 1: Bounding the derivative of the error term.** Under condition (i), the residual has the quadratic form $r(\alpha) = \alpha(1 - \alpha) \cdot \mathbf{r}_2 + O(\epsilon^3)$. This form arises naturally from Taylor expansion when the transition function has bounded second derivatives; the factor $\alpha(1 - \alpha)$ reflects that the residual vanishes at pure-domain endpoints ($\alpha \in \{0, 1\}$).

Taking the derivative with respect to $\alpha$:

$$\frac{dr}{d\alpha} = (1 - 2\alpha) \cdot \mathbf{r}_2 + O(\epsilon^3). \quad (27)$$

Thus:

$$g'(\alpha) = \hat{B}^\dagger \frac{dr}{d\alpha} = (1 - 2\alpha) \cdot \hat{B}^\dagger \mathbf{r}_2 + O(\epsilon^3/\sigma_{\min}). \quad (28)$$

**Step 2: Bounding the magnitude.** Since $\|\hat{B}^\dagger\| = 1/\sigma_{\min}$ and $\|\mathbf{r}_2\| \le C_2\epsilon^2$:

$$\begin{aligned} |g'(\alpha)| &\le |1 - 2\alpha| \cdot \frac{C_2\epsilon^2}{\sigma_{\min}} + O(\epsilon^3/\sigma_{\min}) \\ &\le \frac{C_2\epsilon^2}{\sigma_{\min}} + O(\epsilon^3/\sigma_{\min}). \end{aligned} \quad (29)$$

Under condition (ii), for sufficiently small $\epsilon$:

$$|g'(\alpha)| < \frac{1}{2} < 1. \quad (30)$$

**Step 3: Monotonicity preservation.** The derivative of the recovery map is:

$$\Phi'(\alpha) = 1 + g'(\alpha) > 1 - |g'(\alpha)| > 1 - \frac{1}{2} = \frac{1}{2} > 0. \quad (31)$$

Therefore $\Phi$ is strictly increasing. If $\alpha^*(t)$ is strictly increasing (resp. decreasing), then $\hat{\alpha}(t) = \Phi(\alpha^*(t))$ is also strictly increasing (resp. decreasing). $\square$

**Corollary A.7** (Preservation of Relative Ordering). *Under the conditions of Proposition A.6, for any two time points $t_1 < t_2$:*

$$\alpha^*(t_1) < \alpha^*(t_2) \iff \hat{\alpha}(t_1) < \hat{\alpha}(t_2). \quad (32)$$

*In particular, transition timing, the time at which $\alpha^*(t)$ crosses any threshold $\tau \in (0, 1)$, is preserved up to a small shift bounded by $O(\delta_{\text{approx}}/\sigma_{\min})$.*

*Remark* A.8 (Extension to Multiple Mixing Coefficients). For $K > 2$ domains with $\boldsymbol{\alpha} \in \Delta^{K-1}$, the analysis extends component-wise. If each component $\alpha_k^*(t)$ is monotonic over a time interval and condition (ii) holds, then $\hat{\alpha}_k(t)$ preserves monotonicity on that interval. The relative ordering among components at any fixed time $t$ is preserved when the inter-component gaps exceed the recovery error: $|\alpha_k^*(t) - \alpha_l^*(t)| > 2\delta_{\text{approx}}/\sigma_{\min}$.

*Remark* A.9 (Role of Noise). When observation noise is present, monotonicity holds in expectation but may be violated for individual samples. Theorem 4.3 shows that temporal smoothing reduces the effective noise, restoring monotonicity with high probability when the smoothing window $w$ satisfies $w \gtrsim \sigma^2/(\sigma_{\min}^2 \cdot |\dot{\alpha}^*|^2)$, where $|\dot{\alpha}^*|$ is the rate of change of the true trajectory.

## A.5. Proof of Theorem 4.2: Pointwise Recovery

*Proof.* By Theorem 4.1, the sufficient variability condition ensures that the learned representation satisfies $\hat{\mathbf{z}} = P \cdot h(\mathbf{z})$, where $P$ is a permutation matrix and $h = (h_1, \ldots, h_d)$ is component-wise invertible with each $h_i$ strictly monotonic.

The solver computes mixing coefficients via least-squares projection:

$$\hat{\boldsymbol{\alpha}} = \hat{B}^{\dagger}(\hat{\mathbf{z}}_t - \hat{\boldsymbol{\mu}}^{(0)}), \tag{33}$$

where $\hat{B}^{\dagger} = (\hat{B}^{\top}\hat{B})^{-1}\hat{B}^{\top}$ is the Moore-Penrose pseudoinverse.

Under first-order Taylor expansion of $h$ around $\boldsymbol{\mu}^{(0)}$, the conditional expectation in learned space satisfies:

$$\hat{\boldsymbol{\mu}}_{\text{mixed}}(\boldsymbol{\alpha}) = \hat{\boldsymbol{\mu}}^{(0)} + PD \sum_{k=1}^{K-1} \alpha_k \delta\boldsymbol{\mu}^{(k)} + O(\epsilon^2), \tag{34}$$

where $D = \text{diag}(h_1'(\mu_1^{(0)}), \ldots, h_d'(\mu_d^{(0)}))$ is the diagonal Jacobian. The diagonal structure follows directly from the component-wise nature of $h$ guaranteed by Theorem 4.1. The empirical basis vectors satisfy $\delta\hat{\boldsymbol{\mu}}^{(k)} = PD \cdot \delta\boldsymbol{\mu}^{(k)}$, yielding $\hat{B} = PDB$ where $B = [\delta\boldsymbol{\mu}^{(1)}, \ldots, \delta\boldsymbol{\mu}^{(K-1)}]$ is the basis matrix in the original latent space.

The observation decomposes as $\hat{\mathbf{z}}_t = \hat{\boldsymbol{\mu}}_{\text{mixed}}(\boldsymbol{\alpha}) + \hat{\boldsymbol{\epsilon}}_t$. The conditional expectation under mixed dynamics can be written as:

$$\hat{\boldsymbol{\mu}}_{\text{mixed}}(\boldsymbol{\alpha}) - \hat{\boldsymbol{\mu}}^{(0)} = \hat{B}\boldsymbol{\alpha} + r(\boldsymbol{\alpha}), \tag{35}$$

where $r(\boldsymbol{\alpha}) := \hat{\boldsymbol{\mu}}_{\text{mixed}}(\boldsymbol{\alpha}) - \hat{\boldsymbol{\mu}}^{(0)} - \hat{B}\boldsymbol{\alpha}$ is the first-order approximation residual satisfying $\|r(\boldsymbol{\alpha})\| \leq \delta_{\text{approx}}$ by definition.

Substituting into the solver expression:

$$\hat{\boldsymbol{\alpha}} = \hat{B}^{\dagger}(\hat{B}\boldsymbol{\alpha} + r(\boldsymbol{\alpha}) + \hat{\boldsymbol{\epsilon}}_t) = \hat{B}^{\dagger}\hat{B}\boldsymbol{\alpha} + \hat{B}^{\dagger}r(\boldsymbol{\alpha}) + \hat{B}^{\dagger}\hat{\boldsymbol{\epsilon}}_t. \tag{36}$$

Since $P$ is orthogonal (hence invertible), $D$ is diagonal with nonzero entries (strict monotonicity of each $h_i$ implies $h_i' \neq 0$), and $B$ has full column rank by Assumption 3.1, the product $\hat{B} = PDB$ also has full column rank. To see this, note that $\text{rank}(\hat{B}) = \text{rank}(PDB) = \text{rank}(DB) = \text{rank}(B)$, where the equalities follow from $P$ and $D$ being invertible. The signs of $h_i'$ do not affect this rank preservation. Therefore $\hat{B}^{\dagger}\hat{B} = (\hat{B}^{\top}\hat{B})^{-1}\hat{B}^{\top}\hat{B} = I_{K-1}$, and:

$$\hat{\boldsymbol{\alpha}} - \boldsymbol{\alpha} = \hat{B}^{\dagger}r(\boldsymbol{\alpha}) + \hat{B}^{\dagger}\hat{\boldsymbol{\epsilon}}_t. \tag{37}$$

Taking norms:

$$\|\hat{\boldsymbol{\alpha}} - \boldsymbol{\alpha}\| \leq \|\hat{B}^{\dagger}\| \cdot \|r(\boldsymbol{\alpha})\| + \|\hat{B}^{\dagger}\| \cdot \|\hat{\boldsymbol{\epsilon}}_t\|. \tag{38}$$

The operator norm of the pseudoinverse satisfies $\|\hat{B}^{\dagger}\| = 1/\sigma_{\min}(\hat{B})$. Combined with $\|r(\boldsymbol{\alpha})\| \leq \delta_{\text{approx}}$, this yields the stated bound.

When the first-order approximation is exact ($\delta_{\text{approx}} = 0$) and noise is absent ($\hat{\boldsymbol{\epsilon}}_t = \mathbf{0}$), we have $r(\boldsymbol{\alpha}) = \mathbf{0}$ and thus $\hat{\boldsymbol{\alpha}} = \boldsymbol{\alpha}$. $\qquad\square$

## A.6. Proof of Theorem 4.3: Smooth Trajectory Recovery

We first state the additional assumptions required for this theorem.

**Assumption A.10** (Smooth Trajectory). The true mixing trajectory has bounded total variation:

$$\text{TV}(\boldsymbol{\alpha}^*) := \sum_{t=1}^{T-1} \|\boldsymbol{\alpha}^*(t+1) - \boldsymbol{\alpha}^*(t)\| \leq V. \tag{39}$$

This bound is natural for continuous transitions: smoothly interpolating between two atomic mechanisms over $T$ steps yields $\text{TV} \approx 1$, while erratic switching would have $\text{TV} \approx T$.

**Assumption A.11** (Sub-Gaussian Noise). The observation noise $\hat{\boldsymbol{\epsilon}}_t$ is independent across time with $\mathbb{E}[\hat{\boldsymbol{\epsilon}}_t] = \mathbf{0}$ and sub-Gaussian parameter $\sigma$: for all $\mathbf{v} \in \mathbb{R}^d$, $\mathbb{E}[\exp(\mathbf{v}^{\top}\hat{\boldsymbol{\epsilon}}_t)] \leq \exp(\sigma^2\|\mathbf{v}\|^2/2)$.

*Proof.* **Step 1: Matrix Formulation.** Let $\mathbf{y}_t = \hat{\mathbf{z}}_t - \hat{\boldsymbol{\mu}}^{(0)}$. Stack all time steps:

$$\mathbf{Y} = (\mathbf{y}_1^{\top}, \ldots, \mathbf{y}_T^{\top})^{\top} \in \mathbb{R}^{Td},$$
$$\boldsymbol{\alpha} = (\boldsymbol{\alpha}_1^{\top}, \ldots, \boldsymbol{\alpha}_T^{\top})^{\top} \in \mathbb{R}^{T(K-1)}. \tag{40}$$

Define the block-diagonal design matrix $\mathbf{A} = I_T \otimes \hat{B} \in \mathbb{R}^{Td \times T(K-1)}$ and first-difference matrix $D = D_1 \otimes I_{K-1}$ where

$$D_1 = \begin{pmatrix} -1 & 1 & & \\ & -1 & 1 & \\ & & \ddots & \ddots \end{pmatrix} \in \mathbb{R}^{(T-1) \times T}. \tag{41}$$

The regularized estimator has closed-form solution:

$$\hat{\boldsymbol{\alpha}}^{\text{smooth}} = (\mathbf{A}^{\top}\mathbf{A} + \lambda D^{\top}D)^{-1}\mathbf{A}^{\top}\mathbf{Y} =: M_{\lambda}\mathbf{Y}. \tag{42}$$

**Step 2: Error Decomposition.** The true observation satisfies $\mathbf{Y} = \mathbf{A}\boldsymbol{\alpha}^* + \boldsymbol{\epsilon} + \mathbf{r}$, where $\boldsymbol{\epsilon}$ is stacked noise and $\mathbf{r}$ is the approximation residual with $\|\mathbf{r}\|^2 \leq T\delta_{\text{approx}}^2$. The error decomposes as:

$$\hat{\boldsymbol{\alpha}}^{\text{smooth}} - \boldsymbol{\alpha}^* = \underbrace{(M_{\lambda}\mathbf{A} - I)\boldsymbol{\alpha}^*}_{\text{bias}} + \underbrace{M_{\lambda}\boldsymbol{\epsilon}}_{\text{variance}} + \underbrace{M_{\lambda}\mathbf{r}}_{\text{approximation}}. \tag{43}$$

**Step 3: Bias Bound.** The bias term satisfies:

$$M_{\lambda}\mathbf{A} - I = -\lambda(\mathbf{A}^{\top}\mathbf{A} + \lambda D^{\top}D)^{-1}D^{\top}D. \tag{44}$$

Since $\mathbf{A}^\top \mathbf{A} = I_T \otimes (\hat{B}^\top \hat{B}) \succeq \sigma_{\min}^2 I$, we have $(\mathbf{A}^\top \mathbf{A} + \lambda D^\top D)^{-1} \preceq \sigma_{\min}^{-2} I$. Therefore:

$$\|(M_\lambda \mathbf{A} - I)\boldsymbol{\alpha}^*\| \leq \frac{\lambda}{\sigma_{\min}^2}\|D^\top D \boldsymbol{\alpha}^*\| \leq \frac{\lambda}{\sigma_{\min}^2}\|D\boldsymbol{\alpha}^*\|.$$
(45)

To bound $\|D\boldsymbol{\alpha}^*\|^2$, we use the fact that $\max_t \|\Delta_t\| \leq \mathrm{TV}(\boldsymbol{\alpha}^*) \leq V$:

$$\|D\boldsymbol{\alpha}^*\|^2 = \sum_{t=1}^{T-1} \|\Delta_t\|^2 \leq \max_t \|\Delta_t\| \cdot \sum_t \|\Delta_t\| \leq V \cdot V = V^2.$$
(46)

**Step 4: Variance Bound.** For the variance term, note that $M_\lambda = (\mathbf{A}^\top \mathbf{A} + \lambda D^\top D)^{-1} \mathbf{A}^\top$. Under sub-Gaussian noise:

$$\mathbb{E}\|M_\lambda \boldsymbol{\epsilon}\|^2 = \sigma^2 \mathrm{tr}(M_\lambda M_\lambda^\top).$$
(47)

Since $M_\lambda M_\lambda^\top = (\mathbf{A}^\top \mathbf{A} + \lambda D^\top D)^{-1} \mathbf{A}^\top \mathbf{A} (\mathbf{A}^\top \mathbf{A} + \lambda D^\top D)^{-1}$ and $\mathbf{A}^\top \mathbf{A} \preceq \|\hat{B}\|^2 I$, standard matrix trace bounds yield:

$$\mathrm{tr}(M_\lambda M_\lambda^\top) \leq \frac{T(K-1)}{\sigma_{\min}^2 + \lambda \cdot c_D},$$
(48)

where $c_D > 0$ depends on the structure of $D^\top D$. For $\lambda$ in the relevant range, this simplifies to $O(T(K-1)/(\sigma_{\min}^2 \lambda))$.

**Step 5: Approximation Error.** By the operator norm bound on $M_\lambda$:

$$\|M_\lambda \mathbf{r}\|^2 \leq \|M_\lambda\|^2 \|\mathbf{r}\|^2 \leq \frac{T\delta_{\mathrm{approx}}^2}{\sigma_{\min}^2}.$$
(49)

**Step 6: Combining and Optimizing.** The mean squared error decomposes into bias, variance, and approximation terms. The bias scales as $O(\lambda^2 V^2/T)$ from Step 3, the variance involves the trace of the smoothing matrix from Step 4, and the approximation error contributes $O(\delta_{\mathrm{approx}}^2/\sigma_{\min}^2)$ from Step 5.

For signals with bounded total variation, the bias-variance tradeoff with quadratic smoothing penalties is well-studied (Tibshirani, 2014). The key insight is that the effective degrees of freedom of the smoother scale as $O(T/\lambda^{1/2})$, leading to variance that decreases more slowly than the naive $O(1/\lambda^2)$ rate suggested by pointwise analysis. Balancing these terms yields the optimal regularization $\lambda^* \asymp T^{1/3}$ and the minimax optimal rate:

$$\frac{1}{T}\sum_{t=1}^{T} \mathbb{E}\|\hat{\boldsymbol{\alpha}}_t^{\mathrm{smooth}} - \boldsymbol{\alpha}_t^*\|^2$$
$$= O\left(\frac{(V\sigma)^{2/3}(K-1)^{1/3}}{\sigma_{\min}^{2/3} T^{2/3}} + \frac{\delta_{\mathrm{approx}}^2}{\sigma_{\min}^2}\right).$$
(50)

This $O(T^{-2/3})$ rate matches the minimax rate for nonparametric regression with bounded total variation constraints.

$\square$

*Remark* A.12 (Effect of Simplex Projection). The analysis above is for the unconstrained regularized estimator. Since the true trajectory satisfies $\boldsymbol{\alpha}^*(t) \in \Delta^{K-1}$ for all $t$, the simplex projection in Algorithm 1 can only reduce the $\ell_2$ error to the true trajectory. Therefore, the stated bound remains valid for the projected estimator.

**Efficient Implementation.** The regularized estimator admits $O(TK)$ complexity. When the simplex constraint is relaxed, the problem decouples across the $K-1$ components. For each component $k$, the system reduces to solving $(I + \lambda D_1^\top D_1)\tilde{\alpha}^{(k)} = \tilde{y}^{(k)}$, where $I + \lambda D_1^\top D_1$ is symmetric tridiagonal and solvable in $O(T)$ via the Thomas algorithm. After solving, we project each $\boldsymbol{\alpha}_t$ onto the simplex $\Delta^{K-1}$.

**Selection of Regularization Parameter.** When trajectory smoothness $V$ and noise level $\sigma$ are unknown, generalized cross-validation (GCV) provides automatic selection. In practice, $\lambda \in [0.5, 5]$ works well across diverse settings, with performance relatively insensitive to the exact choice within this range.

### A.7. Proof of Approximation Error Scaling

This section proves the claim in Theorem 4.2 that the first-order approximation error satisfies $\delta_{\mathrm{approx}} = O(\epsilon^2)$, where $\epsilon := \max_k \|W^{(k)} - W^{(0)}\|$ is the maximum perturbation magnitude.

*Proof.* The proof proceeds in three steps, tracking error contributions from both the transition dynamics and the learned representation mapping.

**Step 1: Taylor expansion in original latent space.** Under mixed dynamics, $W_{\mathrm{mixed}} = \sum_k \alpha_k W^{(k)} = W^{(0)} + \sum_{k=1}^{K-1} \alpha_k \delta W^{(k)}$, where $\delta W^{(k)} := W^{(k)} - W^{(0)}$. For smooth activations (e.g., softplus, tanh), we expand $f(\mathbf{z}_{t-1}; W)$ around $W^{(0)}$. For piecewise-linear activations such as LeakyReLU, the expansion holds almost everywhere; the set of non-differentiable points has measure zero under continuous input distributions, and the analysis extends via directional derivatives. Proceeding with the expansion:

$$f(\mathbf{z}_{t-1}; W^{(0)} + \delta W)$$
$$= f(\mathbf{z}_{t-1}; W^{(0)}) + \nabla_W f\big|_{W^{(0)}} \cdot \mathrm{vec}(\delta W)$$
$$+ \frac{1}{2}\mathrm{vec}(\delta W)^\top H_f \, \mathrm{vec}(\delta W) + O(\|\delta W\|^3), \quad (51)$$

where $H_f = \frac{\partial^2 f}{\partial W^2}\big|_{W^{(0)}}$ is the Hessian tensor.

For $\boldsymbol{\alpha} \in \Delta^{K-1}$, we have $\|\delta W\| = \|\sum_k \alpha_k \delta W^{(k)}\| \leq \max_k \|\delta W^{(k)}\| \leq \epsilon$. Define the residual in original space:

$$r_f(\boldsymbol{\alpha}) := \boldsymbol{\mu}_{\text{mixed}}(\boldsymbol{\alpha}) - \boldsymbol{\mu}^{(0)} - \sum_{k=1}^{K-1} \alpha_k \delta \boldsymbol{\mu}^{(k)}. \quad (52)$$

By the Taylor expansion, $\|r_f(\boldsymbol{\alpha})\| \leq C_f \cdot \epsilon^2$ where $C_f = \frac{1}{2}\|H_f\|_{\text{op}}$.

**Step 2: Taylor expansion of the component-wise transformation.** By Theorem 4.1, $\hat{\mathbf{z}} = P \cdot h(\mathbf{z})$ where $h = (h_1, \ldots, h_d)$ is component-wise invertible. Denoting $\boldsymbol{\delta}_\mu := \boldsymbol{\mu}_{\text{mixed}} - \boldsymbol{\mu}^{(0)}$ for brevity and expanding $h$ around $\boldsymbol{\mu}^{(0)}$:

$$h(\boldsymbol{\mu}_{\text{mixed}}) = h(\boldsymbol{\mu}^{(0)}) + D \cdot \boldsymbol{\delta}_\mu + \frac{1}{2} H_h \odot \boldsymbol{\delta}_\mu^{\odot 2} + O(\epsilon^3), \quad (53)$$

where $D = \text{diag}(h_1'(\mu_1^{(0)}), \ldots, h_d'(\mu_d^{(0)}))$ is the diagonal Jacobian and $H_h = \text{diag}(h_1''(\mu_1^{(0)}), \ldots, h_d''(\mu_d^{(0)}))$ captures the second-order terms. The diagonal structure follows directly from the component-wise nature of $h$ guaranteed by Theorem 4.1.

From Step 1, $\boldsymbol{\mu}_{\text{mixed}} - \boldsymbol{\mu}^{(0)} = \sum_k \alpha_k \delta \boldsymbol{\mu}^{(k)} + r_f(\boldsymbol{\alpha})$, which has norm $O(\epsilon)$. Therefore the second-order term from $h$ contributes $O(\epsilon^2)$.

**Step 3: Combining the errors.** In the learned space, the first-order approximation is:

$$\hat{\boldsymbol{\mu}}^{(0)} + \hat{B}\boldsymbol{\alpha} = Ph(\boldsymbol{\mu}^{(0)}) + PD \sum_{k=1}^{K-1} \alpha_k \delta \boldsymbol{\mu}^{(k)}. \quad (54)$$

The true value is $\hat{\boldsymbol{\mu}}_{\text{mixed}}(\boldsymbol{\alpha}) = P \cdot h(\boldsymbol{\mu}_{\text{mixed}}(\boldsymbol{\alpha}))$. The difference consists of two contributions: (1) the transformation $h$ acting on the residual $r_f(\boldsymbol{\alpha})$, contributing $PD \cdot r_f(\boldsymbol{\alpha}) = O(\epsilon^2)$; and (2) the second-order term of $h$, contributing $O(\epsilon^2)$.

Combining these, $\delta_{\text{approx}} \leq C \cdot \epsilon^2$ where the explicit form of $C$ is derived in Section A.8. $\square$

## A.8. Explicit Form of the Approximation Constant

We derive the explicit form of the constant $C$ appearing in Theorem 4.2. The approximation error $\delta_{\text{approx}}$ arises from two sources: the Taylor expansion of the transition function $f$ in the original latent space, and the Taylor expansion of the component-wise transformation $h$ that maps true latents to learned latents.

The two error contributions are:

1. The transformation $h$ acting on $r_f(\boldsymbol{\alpha})$: contributes $\|PD \cdot r_f(\boldsymbol{\alpha})\| \leq \|D\|_{\text{op}} \cdot C_f \cdot \epsilon^2$

2. The second-order term of $h$: contributes $\frac{1}{2}\|H_h\|_\infty \cdot \|\boldsymbol{\mu}_{\text{mixed}} - \boldsymbol{\mu}^{(0)}\|^2 \leq \frac{1}{2}\|H_h\|_\infty \cdot M_\mu^2 \cdot \epsilon^2$

where $M_\mu := \sup_{\boldsymbol{\alpha}} \|\sum_k \alpha_k \delta \boldsymbol{\mu}^{(k)}\|/\epsilon$ is the normalized magnitude of the conditional expectation shift.

Combining these bounds yields the explicit form:

$$\boxed{C = \|D\|_{\text{op}} \cdot \frac{\|H_f\|_{\text{op}}}{2} + \frac{\|H_h\|_\infty \cdot M_\mu^2}{2}}, \quad (55)$$

where:

- $\|D\|_{\text{op}} = \max_i |h_i'(\mu_i^{(0)})|$ is the maximum slope of the component-wise transformation at the baseline

- $\|H_f\|_{\text{op}}$ is the operator norm of the transition function's Hessian with respect to $W$

- $\|H_h\|_\infty = \max_i |h_i''(\mu_i^{(0)})|$ is the maximum curvature of the component-wise transformation

- $M_\mu$ is the normalized magnitude of domain-induced shifts in conditional expectation

**Interpretation.** The constant $C$ captures two distinct sources of nonlinearity: (1) curvature of the transition dynamics in the weight space (via $H_f$), and (2) curvature of the learned representation mapping (via $H_h$). When the learned encoder uses smooth activations (e.g., tanh, softplus) and the transition function has bounded second derivatives, both terms remain finite.

**Empirical Estimation.** While the theoretical expression for $C$ involves quantities that are difficult to compute directly, $\delta_{\text{approx}}$ itself can be estimated empirically when mixed-domain validation data is available. By generating data at known intermediate mixing coefficients $\boldsymbol{\alpha} \in \{0.25, 0.5, 0.75\}$ and comparing $\hat{\boldsymbol{\mu}}_{\text{mixed}}(\boldsymbol{\alpha})$ against the linear prediction $\hat{\boldsymbol{\mu}}^{(0)} + \hat{B}\boldsymbol{\alpha}$, one obtains a direct estimate of the approximation error without needing to compute $C$ explicitly.

## A.9. Analysis of Non-Ideal Conditions

**Effect of Observation Noise.** When observation noise is present, the estimator becomes $\hat{\boldsymbol{\alpha}} = \boldsymbol{\alpha} + \hat{B}^\dagger r(\boldsymbol{\alpha}) + \hat{B}^\dagger \hat{\boldsymbol{\epsilon}}_t$. If $\hat{\boldsymbol{\epsilon}}_t$ is zero-mean with covariance $\Sigma_\epsilon$ and independent of $\boldsymbol{\alpha}$, then:

$$\mathbb{E}[\hat{\boldsymbol{\alpha}}] = \boldsymbol{\alpha} + \hat{B}^\dagger r(\boldsymbol{\alpha}), \quad (56)$$

$$\text{Var}(\hat{\boldsymbol{\alpha}}) = \hat{B}^\dagger \Sigma_\epsilon (\hat{B}^\dagger)^\top. \quad (57)$$

The estimator has bias bounded by $\delta_{\text{approx}}/\sigma_{\min}$, and variance that can be reduced by averaging across $T$ time points: $\text{Var}(\bar{\boldsymbol{\alpha}}) = \frac{1}{T}\hat{B}^\dagger \Sigma_\epsilon (\hat{B}^\dagger)^\top$.

**Effect of Basis Estimation Error.** In practice, the empirical basis matrix may contain estimation error: $\hat{B}_{\text{emp}} = \hat{B} + E$. When $\|E\|$ is small relative to $\sigma_{\min}$, first-order perturbation analysis gives:

$$\hat{\boldsymbol{\alpha}} \approx (I - \hat{B}^{\dagger} E)\boldsymbol{\alpha} + \hat{B}^{\dagger} r(\boldsymbol{\alpha}) + \hat{B}^{\dagger}\hat{\boldsymbol{\epsilon}}_t + O(\|E\|^2). \quad (58)$$

This analysis shows that basis estimation error introduces a linear transformation $M = I - \hat{B}^{\dagger} E$ between true and recovered coefficients. When ground-truth boundary conditions are available (i.e., trajectories with known one-hot $\boldsymbol{\alpha}$ values at endpoints), $M$ can be estimated and inverted for calibration as described in Remark 4.4.

### A.10. Summary of Assumptions

For reference, we summarize all assumptions required for the theoretical results. We use the notation $\delta W^{(k)} := W^{(k)} - W^{(0)}$ throughout, treating domain 0 as baseline.

**A1** (Invertible Mixing) The observation function $g : \mathbb{R}^d \to \mathbb{R}^p$ is invertible.

**A2** (Conditional Independence) The learned representation $\hat{\mathbf{z}}_t$ has mutually independent components conditional on $\hat{\mathbf{z}}_{t-1}$.

**A3** (Sufficient Variability) The function vectors $\{\mathbf{s}_{k,t}, \dot{\mathbf{s}}_{k,t}\}_{k=1}^d$ derived from the conditional log-densities are linearly independent.

**A3'** (Row-wise Non-degeneracy) For each component $k \in \{1, \ldots, d\}$, the row vectors $\{(\delta W^{(u)})_{k,:}\}_{u=1}^{K-1}$ are linearly independent in $\mathbb{R}^d$. This is a sufficient condition for A3 (via Lemma A.3) and implies the capacity constraint $K \le d + 1$.

**A4** (Basis Full Rank) The basis matrix $\hat{B}$ has full column rank, i.e., $\sigma_{\min}(\hat{B}) > 0$.

**A5** (Smoothness) The transition function $f$ is twice continuously differentiable in $W$, and the component-wise transformation $h$ is twice continuously differentiable in a neighborhood of $\boldsymbol{\mu}^{(0)}$.

**A6** (Trajectory Regularity) For Theorem 4.3: the true trajectory has bounded total variation $\text{TV}(\boldsymbol{\alpha}^*) \le V$, and noise is i.i.d. sub-Gaussian with parameter $\sigma$.

Assumptions A1–A3 are standard in temporal causal representation learning (Yao et al., 2022a). Assumption A3' provides a verifiable sufficient condition for A3 in our setting; it is satisfied generically when mechanism perturbations are drawn from continuous distributions. Assumptions A4–A5 are mild regularity conditions. A6 is required only for the improved $O(T^{-2/3})$ rate in Theorem 4.3.

**Correspondence with Main Text.** Assumption 3.1 (Distinguishable Mechanisms) in Section 3 states that $\{W^{(k)} - W^{(0)}\}_{k=1}^{K-1}$ are linearly independent. Combined with row-wise non-degeneracy (A3'), this implies A3 via Lemma A.3, and directly implies A4.

## B. Algorithm

---
**Algorithm 1** Mechanism Trajectory Inference
---
1: **Input:** Trained encoder $g_\phi^{-1}$, basis matrix $\hat{B}$, baseline mean $\hat{\boldsymbol{\mu}}^{(0)}$, test trajectory $\{\mathbf{x}_t\}_{t=1}^T$, window size $w$
2: **Output:** Mixing coefficients $\{\bar{\boldsymbol{\alpha}}(t)\}_{t=1}^T$
3: Compute pseudoinverse $\hat{B}^{\dagger} \leftarrow (\hat{B}^{\top}\hat{B})^{-1}\hat{B}^{\top}$
4: **for** $t = L + 1$ to $T$ **do**
5:     $\mathbf{z}_t \leftarrow \boldsymbol{\mu}_t$ from $g_\phi^{-1}(\mathbf{x}_t, \mathbf{x}_{t-L:t-1})$ *// posterior mean*
6:     $\hat{\boldsymbol{\alpha}}(t) \leftarrow \text{Proj}_{\Delta^{K-1}}\left(\hat{B}^{\dagger}(\mathbf{z}_t - \hat{\boldsymbol{\mu}}^{(0)})\right)$ *// Theorem 4.2*
7: **end for**
8: *// Temporal smoothing (Theorem 4.3)*
9: **for** $t = L + 1 + w$ to $T - w$ **do**
10:     $\bar{\boldsymbol{\alpha}}(t) \leftarrow \frac{1}{2w+1}\sum_{s=t-w}^{t+w}\hat{\boldsymbol{\alpha}}(s)$
11: **end for**
12: **return** $\{\bar{\boldsymbol{\alpha}}(t)\}$ *// Boundary points use unsmoothed $\hat{\boldsymbol{\alpha}}$*
---

## C. Baseline Methods

We compare TRACE against the following temporal causal representation learning methods:

**TDRL (Yao et al., 2022a).** Temporally Disentangled Representation Learning recovers latent causal variables by exploiting time-delayed dependencies. It assumes that latent variables follow a first-order Markov process with domain-specific transition distributions, achieving identifiability through sufficient variability across discrete domains.

**NCTRL (Song et al., 2023).** Nonstationary Causal Temporal Representation Learning extends TDRL to handle unknown domain labels by introducing a learnable gating mechanism. We evaluate two variants: NCTRL-hard uses discrete one-hot gating, while NCTRL-soft allows probabilistic routing. Both assume discrete mechanism switches rather than continuous transitions.

**LEAP (Yao et al., 2022b).** Latent Causal Process Recovery learns temporally causal representations by enforcing sparsity constraints on the transition function. It requires known domain labels and assumes discrete mechanism changes across domains.

**iVAE (Khemakhem et al., 2020).** Identifiable Variational Autoencoder achieves nonlinear ICA identifiability by con-

ditioning on auxiliary variables (e.g., domain labels). While not specifically designed for temporal data, it provides a strong baseline for identifiable representation learning with discrete auxiliary information.

**PCL (Hyvarinen & Morioka, 2017).** Permutation Contrastive Learning achieves nonlinear ICA identifiability by exploiting temporal dependencies in stationary time series. It learns to discriminate true temporal sequences from permuted ones, recovering independent components up to permutation and component-wise transformation.

All baselines assume either fixed causal mechanisms or discrete switches between mechanisms. In contrast, TRACE explicitly models continuous mechanism transitions as convex combinations of atomic mechanisms.

**Methods Not Compared.** The following recent methods address settings orthogonal to ours and are therefore not included as baselines.

**IDOL (Li et al., 2024).** IDOL identifies temporally causal representations with instantaneous dependencies by imposing sparse influence constraints. It targets settings where latent variables have same-timestep causal effects. Our formulation explicitly assumes no instantaneous effects (Eq. 1), making IDOL's machinery unnecessary for our setting.

**CaRiNG (Chen et al., 2024).** CaRiNG extends temporal causal representation learning to non-invertible generation processes, addressing scenarios with information loss such as 3D-to-2D projection. Our framework assumes invertible mixing (Assumption in Theorem 4.1), a standard setting where CaRiNG's relaxation is not required.

**CtrlNS (Song et al., 2024).** CtrlNS identifies discrete regimes without domain labels by assuming *sparse transitions*, where mechanisms remain stable most of the time with only occasional discrete switches. This sparsity assumption is fundamentally incompatible with our continuous transition setting, where $\boldsymbol{\alpha}(t)$ evolves at every timestep. Applying CtrlNS to such data would violate its identifiability conditions.

**CHiLD (Li et al., 2025b).** CHiLD addresses hierarchical latent dynamics where multi-layer latent variables exhibit temporal dependencies across different levels of abstraction. Our framework assumes a single layer of latent causal variables, a setting where hierarchical modeling is not applicable.

## D. Experimental Details

This section provides implementation details for all experiments in Section 6.

### D.1. Synthetic Data Generation

We define a base transition matrix $W_{\text{base}} \in \mathbb{R}^{8 \times 8}$ and construct $K = 5$ atomic mechanisms with $W^{(k)} = W_{\text{base}} + \delta W^{(k)}$, where each $\delta W^{(k)}$ modifies a distinct off-diagonal entry. This ensures linear independence per Assumption 3.1.

Latent variables evolve according to second-order Markov dynamics:

$$\mathbf{z}_t = \sigma\left(\sigma(\mathbf{z}_{t-1}W^{(k)}) + \sigma(\mathbf{z}_{t-2}W_{\text{lag2}})\right) + \boldsymbol{\epsilon}_t, \quad (59)$$

where $\sigma$ is LeakyReLU with negative slope 0.2 and $\boldsymbol{\epsilon}_t \sim \mathcal{N}(\mathbf{0}, 0.1^2\mathbf{I})$. High-dimensional observations $\mathbf{x}_t \in \mathbb{R}^{16}$ are generated via an orthogonal MLP ensuring invertibility.

For evaluation, mixed trajectories use time-varying transition matrices:

$$W_{\text{mixed}}(t) = W_{\text{base}} + \sum_{k=1}^{K} w_k(t) \cdot \delta W^{(k)}, \quad (60)$$

with $w_k(t)$ following linear or sinusoidal interpolation schedules.

**Training Details.** We train on 40,000 trajectories per domain (200,000 total), each of length 50. The encoder is a 3-layer MLP with hidden dimension 128 and LeakyReLU activations. Each expert is implemented as a 2-layer masked autoregressive flow. We use Adam optimizer with learning rate $10^{-4}$ and train for 100 epochs with batch size 256.

### D.2. CartPole Environment

We construct a CartPole-inspired dynamical system with latent state $\mathbf{z}_t = (x, v, \theta, \omega)^{\top} \in \mathbb{R}^4$ representing cart position, velocity, pole angle, and angular velocity.

**Dynamics.** The system follows second-order Markov dynamics:

$$\mathbf{z}_t = \sigma(\sigma(\mathbf{z}_{t-1}W^{(k)}) + \sigma(\mathbf{z}_{t-2}W_{\text{lag2}})) + \boldsymbol{\epsilon}_t, \quad (61)$$

where $\sigma$ is LeakyReLU and $\boldsymbol{\epsilon}_t \sim \mathcal{N}(\mathbf{0}, 0.05^2\mathbf{I})$.

**Domain Construction.** We define five domains ($k \in \{0, 1, 2, 3, 4\}$) with $W^{(k)} = W_{\text{base}} + \delta W^{(k)}$. Each $\delta W^{(k)}$ modifies a single off-diagonal edge in the causal graph:

- Domain 0: Baseline (no perturbation)
- Domain 1: Enhanced $v \to \theta$ coupling (velocity affects angle)

- Domain 2: Enhanced $\theta \to \omega$ coupling (angle affects angular velocity)

- Domain 3: Enhanced $\omega \to v$ coupling (angular velocity affects cart velocity)

- Domain 4: Enhanced $x \to \omega$ coupling (position affects angular velocity)

**Observation Generation.** Observations are $128 \times 128$ grayscale images rendered from the latent state using a deterministic rendering function that maps $(x, \theta)$ to cart and pole positions.

**Training Details.** We train on 1,000 trajectories per domain (5,000 total), each of length 100. The encoder uses a CNN with architecture: Conv(32, 4, 2) $\to$ Conv(64, 4, 2) $\to$ Conv(128, 4, 2) $\to$ FC(256) $\to$ FC(4). We train for 200 epochs with batch size 64.

**D.3. Vehicle Turning (UAVDT)**

We use the UAVDT dataset (Du et al., 2018), which provides UAV-captured urban traffic scenes with annotated vehicle bounding boxes.

**Data Selection.** We extract vehicle trajectories at major intersections where turning maneuvers are observable. Selection criteria:

- Trajectory length $\geq 20$ frames

- Total direction change $\geq 60°$

- Continuous tracking (no occlusion gaps $> 3$ frames)

**Domain Definition.** We define two atomic mechanisms:

- Domain 0 (Horizontal): Vehicles moving primarily in the $x$-direction (left-right)

- Domain 1 (Vertical): Vehicles moving primarily in the $y$-direction (up-down)

Pure-domain training data is collected from straight-moving vehicles in each direction.

**Proxy Construction.** Since ground-truth mechanism labels are unavailable, we construct a physically-motivated proxy from annotated bounding box centers $\mathbf{p}_t$. We compute instantaneous velocity and direction:

$$\mathbf{v}_t = \mathbf{p}_{t+1} - \mathbf{p}_t, \tag{62}$$
$$\theta_t = \text{atan2}(v_y, v_x). \tag{63}$$

The proxy mixing coefficients are defined as:

$$\alpha_{\text{horizontal}}^{\text{proxy}} = \frac{|\cos(\theta_t)|}{|\cos(\theta_t)| + |\sin(\theta_t)|}, \tag{64}$$

$$\alpha_{\text{vertical}}^{\text{proxy}} = \frac{|\sin(\theta_t)|}{|\cos(\theta_t)| + |\sin(\theta_t)|}. \tag{65}$$

This proxy reflects the intuition that vehicle dynamics are dominated by horizontal mechanisms (lateral steering) when moving horizontally, and vertical mechanisms (longitudinal acceleration) when moving vertically.

**Proxy Limitations.** This proxy conflates kinematic state (velocity direction) with dynamic mechanism (causal relationships). A vehicle at $\theta = 45°$ is not necessarily in a "50%-50%" mechanism state, since the actual mechanism depends on steering input, road conditions, and vehicle dynamics not captured by velocity alone. The proxy is best interpreted as a behavioral correlate rather than a direct measurement of mechanism state. Our results demonstrate that TRACE recovers trajectories correlating strongly with this proxy (0.96), with smooth transitions whose qualitative structure aligns with physical expectations.

**Training Details.** We train on 500 pure-domain trajectories per direction. The encoder uses a CNN similar to CartPole but with temporal convolutions to handle the 5-frame input. We train for 150 epochs.

**D.4. Human Motion Capture (CMU)**

We use the CMU Motion Capture dataset, which contains skeletal motion data recorded at 120 Hz.

**Data Selection.** We select walk-to-run transition trials from subjects 2, 7, 8, 9, 16, and 35, totaling over 1,000 frames of transition data. Pure walking and running sequences are collected from separate trials of the same subjects.

**Feature Extraction.** From the 31-joint skeleton, we extract:

- Joint positions (93 dimensions)

- Joint velocities (93 dimensions)

- Hip height and speed (2 dimensions)

The observation dimension is $p = 188$ after concatenation.

**Proxy Construction.** Since ground-truth mechanism labels are unavailable, we use normalized hip speed as a proxy:

$$\alpha_{\text{run}}^{\text{proxy}}(t) = \frac{v_{\text{hip}}(t) - v_{\text{walk}}}{v_{\text{run}} - v_{\text{walk}}}, \tag{66}$$

where $v_{\text{walk}}$ and $v_{\text{run}}$ are mean hip speeds during pure walking and running. This proxy assumes a monotonic relationship between locomotion speed and gait mechanism, supported by biomechanics research showing that gait transitions occur at characteristic speeds determined by energetic optimality (Hreljac, 1993).

**Proxy Limitations.** The walk-run transition is not strictly linear in speed; there exists a hysteresis region where both gaits are energetically viable (Diedrich & Warren Jr, 1995).Additionally, hip speed captures only one aspect of gait; stride frequency, duty factor, and joint coordination patterns may provide complementary information. Our results demonstrate strong correlation (0.86) between TRACE's recovered trajectories and this proxy, with smooth transitions aligning with physical expectations.

**Training Details.** We train on 2,000 frames of pure walking and 2,000 frames of pure running. The encoder is a 4-layer MLP with hidden dimension 256. We use $d = 8$ latent dimensions and lag $L = 3$. Training runs for 100 epochs with batch size 128.

### D.5. Implementation Details

**Model Configuration.** Table 6 summarizes the key hyperparameters. For synthetic experiments, we use $K = 5$ or $K = 10$ domains with latent dimension $d = 8$ and lag $L = 2$. The encoder and decoder are 3-layer MLPs with hidden dimension 128. Each expert is implemented as a nonparametric transition prior following the normalizing flow formulation.

*Table 6.* Hyperparameters for synthetic experiments.

| Hyperparameter | $K = 5$ | $K = 10$ |
|---|---|---|
| Latent dimension $d$ | 8 | 8 |
| Lag $L$ | 2 | 2 |
| Embedding dimension | 2 | 8 |
| Encoder hidden dim | 128 | 128 |
| Decoder hidden dim | 128 | 128 |
| Learning rate | $5 \times 10^{-4}$ | $5 \times 10^{-4}$ |
| $\beta$ (KL weight) | $2 \times 10^{-3}$ | $2 \times 10^{-3}$ |
| $\gamma$ (prior weight) | $2 \times 10^{-2}$ | $2 \times 10^{-2}$ |
| Batch size | 64 | 64 |
| Epochs | 100 | 100 |

**Computational Resources.** All experiments were conducted on NVIDIA A100 GPUs. Training times are reported in Table 7.

**Software.** The implementation uses PyTorch 2.0 with CUDA 11.8. Code and pretrained models will be released upon publication.

*Table 7.* Training time by experiment type.

| Experiment | GPUs | Training Time |
|---|---|---|
| Synthetic ($K = 5$) | $1 \times$A100 | 6–8 hours |
| Synthetic ($K = 10$) | $2 \times$A100 | 8–10 hours |
| CartPole (image) | $2 \times$A100 | 15–20 hours |
| Vehicle (image) | $2 \times$A100 | 15–20 hours |
| MoCap (skeleton) | $1 \times$A100 | 4–6 hours |

## E. Extended Results

### E.1. Extended Theoretical Validation

Table 8 reports all quantities appearing in the error bound of Theorem 4.2. Key observations: (1) $\sigma_{\min}$ increases with perturbation magnitude $\delta$, confirming that larger domain differences improve distinguishability; (2) $\delta_{\text{approx}}$ remains stable around 0.1, validating the linear interpolation assumption for piecewise-linear transitions; (3) $\text{SNR}_{\text{eff}}$ consistently predicts recovery quality across all configurations.

*Table 8.* Extended SNR validation results. All quantities correspond to those in Theorem 4.2.

| $\sigma_\epsilon$ | $\|W^{(k)} - W^{(0)}\|$ | MCC | Corr | $\sigma_{\min}$ | $\|\hat{\epsilon}\|$ | $\delta_{\text{ap}}$ | SNR |
|---|---|---|---|---|---|---|---|
| **Scheme 1: Vary Noise** (fixed $\|W^{(k)} - W^{(0)}\| = 0.5$) | | | | | | | |
| 0.01 | 0.5 | **.996** | **.998** | .830 | 1.919 | .100 | **.411** |
| 0.05 | 0.5 | .994 | .998 | .729 | 1.779 | .100 | .388 |
| 0.10 | 0.5 | .984 | .998 | .482 | 1.820 | .100 | .251 |
| 0.20 | 0.5 | .849 | .996 | .207 | 1.950 | .100 | .101 |
| 0.50 | 0.5 | .703 | .854 | .150 | 2.400 | .100 | .060 |
| **Scheme 2: Vary** $\delta$ (fixed $\sigma_\epsilon = 0.1$) | | | | | | | |
| 0.10 | 0.1 | .934 | .972 | .066 | 1.435 | .100 | .043 |
| 0.10 | 0.2 | .953 | .996 | .076 | 1.517 | .100 | .047 |
| 0.10 | 0.3 | .970 | .997 | .157 | 1.588 | .100 | .093 |
| 0.10 | 0.5 | .984 | .998 | .482 | 1.820 | .100 | .251 |
| 0.10 | 0.7 | **.990** | **.998** | .674 | 2.060 | .100 | **.312** |

### E.2. Additional Calibration Results

Figure 8 illustrates scale calibration on a simpler two-domain transition. Raw predictions preserve trajectory shape but exhibit scale compression; two-point calibration using boundary conditions recovers ground truth with MSE $= 0.0005$.

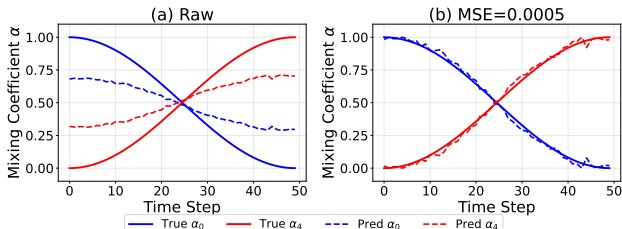

*Figure 8.* Scale calibration on a two-domain transition. **Left:** Raw predictions preserve shape but exhibit scale compression. **Right:** Two-point calibration recovers ground truth (MSE $= 0.0005$).

The calibration procedure works as follows. Given boundary conditions corresponding to known pure-domain endpoints, we fit an affine transformation:

$$\boldsymbol{\alpha}^{\text{cal}} = a \cdot \hat{\boldsymbol{\alpha}} + b, \tag{67}$$

where $a, b$ are determined by matching the boundary values. This simple two-point calibration is sufficient because the scale distortion is approximately constant along the trajectory, as predicted by the linear relationship in the proof of Theorem 4.2.

### E.3. Assumption Verification Protocol

The convex interpolation assumption underlying our framework posits that intermediate mechanisms are expressible as weighted combinations of atomic mechanisms. We provide a statistical testing protocol for verification.

**Step 1: Compute Pure-Domain Residuals.** For each pure-domain validation sample $(\mathbf{x}_t, k)$ where $k$ is the known domain label, compute:

$$r_t^{\text{pure}} = \|\tilde{\mathbf{z}}_t - \hat{\boldsymbol{\mu}}^{(k)}\|_2, \tag{68}$$

where $\tilde{\mathbf{z}}_t$ is the aligned latent representation and $\hat{\boldsymbol{\mu}}^{(k)}$ is the estimated conditional mean for domain $k$. Under correct model specification, $r_t^{\text{pure}}$ reflects observation noise only.

**Step 2: Compute Transition Residuals.** For each transition sample $\mathbf{x}_t$, first recover mixing coefficients $\hat{\boldsymbol{\alpha}}_t$ via Algorithm 1, then compute:

$$r_t^{\text{trans}} = \|\tilde{\mathbf{z}}_t - \hat{\boldsymbol{\mu}}^{(0)} - \hat{B}\hat{\boldsymbol{\alpha}}_t\|_2. \tag{69}$$

Under valid convex interpolation, this residual should also reflect observation noise only.

**Step 3: Statistical Test.** Apply a two-sample Kolmogorov-Smirnov (KS) test:

$$H_0 : F_{r^{\text{pure}}} = F_{r^{\text{trans}}} \quad \text{vs.} \quad H_1 : F_{r^{\text{pure}}} \neq F_{r^{\text{trans}}}, \tag{70}$$

where $F$ denotes the cumulative distribution function. A $p$-value exceeding 0.05 indicates insufficient evidence to reject the convex interpolation assumption.

**Interpretation Guidelines.**

- $p > 0.05$: Assumption is supported; recovered trajectories may be interpreted with confidence.

- $0.01 < p \leq 0.05$: Marginal; interpret results with caution, consider examining residual time series for localized violations.

- $p \leq 0.01$: Assumption likely violated; consider increasing $K$ or restricting analysis to trajectory segments with low residuals.

**Results on Experimental Datasets.** We applied this protocol to all datasets (synthetic, CartPole, Vehicle, MoCap); in each case, the KS test yielded $p > 0.05$, supporting the validity of convex interpolation.

**Detecting Assumption Violations: Synthetic Demonstration.** To validate that our protocol detects genuine violations, we construct a synthetic counter-example where the convex assumption is intentionally violated. We generate data with three atomic mechanisms ($K = 3$), but introduce an *emergent edge* during transitions: when $0.3 < \alpha_1 < 0.7$, an additional causal edge ($z_2 \rightarrow z_5$) activates that is absent in all pure domains.

The transition residuals exhibit a clear spike during the emergent-edge region, and the KS test strongly rejects $H_0$ ($p < 0.001$). This demonstrates that our protocol successfully identifies assumption violations.

### E.4. Extended Ablation Studies

We provide comprehensive ablation experiments examining factors affecting recovery performance: temporal smoothing, the number of active domains $K_{\text{active}}$, trajectory complexity, and model capacity $K_{\text{total}}$.

#### E.4.1. TRAJECTORY COMPLEXITY DEFINITIONS

We define three levels of trajectory complexity to systematically evaluate recovery difficulty:

**Simple (Sequential) Trajectories.** Linear sequential transitions where at most two domains have non-zero mixing coefficients at any time. Each active domain reaches $\alpha = 1.0$ in sequence before transitioning to the next.

**Medium (Overlapping) Trajectories.** Gaussian-shaped activations where multiple domains can be simultaneously active with overlapping support. Each domain has a clear peak moment (achieving $\alpha \approx 0.6$–$0.8$) but transitions smoothly with neighboring domains.

**Complex (Oscillating) Trajectories.** High-frequency cosine superposition where all active domains contribute throughout the trajectory. No domain achieves dominance ($\max \alpha \approx 0.4$–$0.5$), and coefficients oscillate non-monotonically:

$$\alpha_k(t) \propto \frac{1}{2}\left(1 + \cos\left((1 + 0.5k) \cdot 2\pi \frac{t}{T} + \frac{k\pi}{K_{\text{active}}}\right)\right). \tag{71}$$

#### E.4.2. TEMPORAL SMOOTHING WINDOW SIZE

Table 10 reports recovery performance under varying window sizes $w$ on synthetic data with $K_{\text{active}} = 5$ and com-

Table 9. Properties of trajectory complexity levels.

| Property | Simple | Medium | Complex |
|---|---|---|---|
| Max simultaneous domains | 2 | 2–3 | All |
| Peak $\alpha$ value | 1.0 | 0.7 | 0.5 |
| Monotonic segments | Yes | Yes | No |

plex trajectories. Results confirm Theorem 4.3: temporal smoothing substantially improves recovery by averaging out observation noise.

Table 10. Effect of smoothing window $w$ ($K_{\text{active}} = 5$, complex trajectory).

| $w$ | Corr. ↑ | MAE ↓ |
|---|---|---|
| 0 (none) | $0.575 \pm 0.023$ | $0.106 \pm 0.004$ |
| 3 | $0.859 \pm 0.022$ | $0.057 \pm 0.002$ |
| 5 (default) | $0.890 \pm 0.018$ | $0.050 \pm 0.004$ |
| 7 | $\mathbf{0.900 \pm 0.017}$ | $\mathbf{0.050 \pm 0.004}$ |
| 10 | $0.876 \pm 0.014$ | $0.059 \pm 0.003$ |

Without smoothing ($w = 0$), pointwise estimates exhibit high variance with correlation of only 0.575. Moderate smoothing ($w = 5$–$7$) achieves optimal performance, improving correlation by 0.32 absolute points. Excessive smoothing ($w = 10$) slightly degrades performance by over-smoothing rapid transitions. We use $w = 5$ as the default throughout all experiments.

### E.4.3. MODEL CAPACITY ($K_{\text{TOTAL}}$)

We compare models trained with $K_{\text{total}} = 5$ versus $K_{\text{total}} = 10$ on identical recovery tasks. Table 11 summarizes results across trajectory types.

Table 11. Effect of model capacity $K_{\text{total}}$ on recovery (correlation ↑).

| $K_{\text{active}}$ | $K_{\text{total}} = 5$ | | | $K_{\text{total}} = 10$ | | |
|---|---|---|---|---|---|---|
| | Simp. | Med. | Comp. | Simp. | Med. | Comp. |
| 2 | .981 | .990 | .975 | **.993** | **.997** | **.979** |
| 3 | .986 | .984 | .953 | **.989** | **.989** | **.971** |
| 4 | .984 | .974 | .903 | **.988** | **.982** | **.919** |
| 5 | **.980** | **.970** | .810 | .979 | .965 | **.835** |

The larger model ($K_{\text{total}} = 10$) consistently matches or outperforms the smaller model for $K_{\text{active}} \leq 4$. This contradicts the naive hypothesis that more experts would dilute encoder capacity. The explanation lies in the identifiability theory: training on more domains provides greater distributional variation across the dataset, which strengthens the sufficient variability condition required by Theorem 4.1. A stronger variability signal enables the encoder to learn more accurate latent representations $\hat{\mathbf{z}}_t$, which in turn improves downstream trajectory recovery regardless of how many domains are active at inference time.

### E.4.4. FULL SCALING BEHAVIOR

We examine complete scaling behavior by varying $K_{\text{active}}$ from 2 to 10 with the $K_{\text{total}} = 10$ model. Table 12 reveals distinct performance regimes.

**Performance Regimes.** The results reveal four distinct regimes:

*Regime I ($K_{active} \leq 5$): Excellent.* All trajectory types achieve correlation above 0.83. Simple/medium trajectories maintain >0.96.

*Regime II ($K_{active} = 6$): Good.* Simple/medium remain strong (0.957–0.968), but complex drops to 0.692.

*Regime III ($K_{active} = 7$): Degradation onset.* A sharp phase transition: simple drops from 0.968 to 0.776; variance increases substantially.

*Regime IV ($K_{active} \geq 8$): Failure.* Correlation falls below 0.5 for all types. At $K_{\text{active}} = 10$, complex trajectory correlation approaches zero (0.041).

**Geometric Interpretation.** The phase transition at $K_{\text{active}} = 7$ has a geometric explanation rooted in the distinction between linear independence and orthogonality. Our identifiability theory (Assumption 3.1) requires the basis vectors $\{\delta\hat{\boldsymbol{\mu}}^{(k)}\}$ to be *linearly independent*, but not necessarily orthogonal.

When $K_{\text{active}}$ is small (e.g., $K_{\text{active}} = 3$), two basis vectors in an 8-dimensional space can easily maintain near-orthogonality, yielding large pairwise angles and a well-conditioned basis matrix with large $\sigma_{\min}$. As $K_{\text{active}}$ increases, fitting ($K_{\text{active}} - 1$) vectors into $d$ dimensions forces them geometrically closer together. While technically remaining linearly independent, their pairwise angles shrink, and some vectors become nearly expressible as linear combinations of others.

This geometric crowding drives $\sigma_{\min}(\hat{B}) \to 0$, directly inflating the error bound $\|\hat{\boldsymbol{\alpha}} - \boldsymbol{\alpha}\| \leq \|\hat{\boldsymbol{\epsilon}}\|/\sigma_{\min}$ from Theorem 4.2. Empirically, we observe $\sigma_{\min} = 0.48$ at $K_{\text{active}} = 5$ but $\sigma_{\min} = 0.09$ at $K_{\text{active}} = 7$, a $5\times$ degradation that directly explains the correlation drop.

The constraint $K_{\text{active}} \lesssim d$ is therefore not a limitation of TRACE specifically, but a fundamental geometric fact: recovering ($K_{\text{active}} - 1$) mixing coefficients requires ($K_{\text{active}} - 1$) distinguishable directions in a $d$-dimensional space.

### E.4.5. SUMMARY AND PRACTICAL RECOMMENDATIONS

**Practical Recommendations.** **(1)** TRACE works reliably for $K_{\text{active}} \leq d - 2$ with sequential or overlapping transitions (correlation >0.95). **(2)** These conditions match real-world systems: vehicle turning (2 regimes), gait transitions (2–

*Table 12.* Full scaling results ($K_{\text{total}} = 10$, $d = 8$). $\boldsymbol{\alpha}$ recovery degrades with $K_{\text{active}}$ while $\mathbf{W}$ recovery remains stable, confirming the geometric bottleneck affects inference-time decomposition rather than mechanism learning.

| | $\boldsymbol{\alpha}$ recovery (Corr. ↑) | | | $\mathbf{W}$ recovery (Corr. ↑) | | |
|---|---|---|---|---|---|---|
| $K_{\text{active}}$ | Simp. | Med. | Comp. | Simp. | Med. | Comp. |
| 2 | .993±.004 | .997±.002 | .979±.003 | 1.000±.000 | 1.000±.000 | 1.000±.000 |
| 3 | .989±.004 | .989±.006 | .971±.009 | 1.000±.000 | 1.000±.000 | 1.000±.000 |
| 4 | .988±.003 | .982±.004 | .919±.016 | .999±.000 | .999±.000 | .999±.000 |
| 5 | .979±.007 | .965±.014 | .835±.035 | .999±.000 | .999±.000 | .999±.000 |
| 6 | .968±.009 | .957±.012 | .692±.050 | .998±.000 | .999±.000 | .999±.000 |
| 7 | .776±.053 | .735±.061 | .459±.052 | .995±.000 | .996±.000 | .998±.000 |
| 8 | .462±.084 | .389±.091 | .136±.094 | .988±.000 | .992±.000 | .995±.000 |
| 9 | .289±.089 | .222±.112 | .046±.057 | .988±.000 | .993±.001 | .996±.000 |
| 10 | .304±.060 | .252±.090 | .041±.036 | .989±.000 | .993±.000 | .997±.000 |

*Table 13.* Summary of factors affecting recovery performance.

| Factor | Impact | Recommendation |
|---|---|---|
| Trajectory complexity | High | Prefer sequential transitions |
| $K_{\text{active}}$ | High | Keep $\leq d - 2$ (here $\leq 6$) |
| Smoothing window $w$ | Medium | Use $w = 5$–$7$ |
| Model capacity $K_{\text{total}}$ | Low | Larger is fine |

3). **(3)** For systems requiring more simultaneous regimes, increase the latent dimension $d$ or consider hierarchical decomposition. **(4)** The correlation gap between simple and complex trajectories (0.14–0.27) suggests prioritizing data collection during structured, sequential transitions when possible.

### E.5. OOD Generalization

We evaluate the importance of Stage 2 (least-squares projection) for out-of-distribution (OOD) generalization to unseen mechanism transitions.

**Setup.** We train on pure-domain data from domains $\{0, 1, 2, 3, 4\}$ and evaluate on three-domain transition trajectories ($0 \rightarrow 2 \rightarrow 4$) that were never observed during training. We compare: (1) Stage 1 only, which uses the learned gating network to predict $\boldsymbol{\alpha}$; and (2) the full two-stage model, which applies Algorithm 1 for trajectory recovery.

**Results.** Figure 9 and Table 14 demonstrate that Stage 2 is essential for OOD generalization. Stage 1 alone achieves only 0.313 correlation on OOD transitions (despite 0.990 on in-distribution data), as the gating network overfits to pure-domain inputs. The full model recovers OOD trajectories with 0.945 correlation, confirming that the least-squares projection generalizes to unseen mechanism combinations.

**Optional: Distribution Alignment.** In scenarios where the observation distribution shifts significantly between pure-domain training data and transition test data, an optional affine adapter can be applied. The adapter learns a

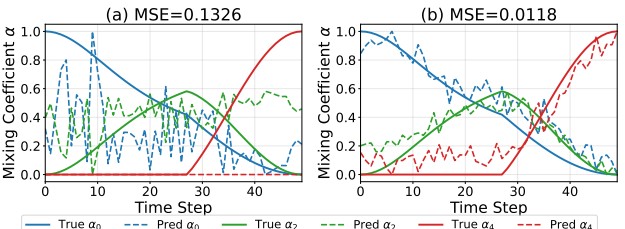

*Figure 9.* OOD weight recovery on a three-domain transition ($0 \rightarrow 2 \rightarrow 4$). **(a)** Stage 1 only: gating network fails to generalize. **(b)** Full model: least-squares projection recovers the trajectory accurately.

*Table 14.* Ablation on two-stage training. Stage 2 is essential for OOD generalization.

| Eval Data | Configuration | Weight Corr. ↑ | MSE ↓ |
|---|---|---|---|
| ID | Stage 1 only | 0.990 | 0.014 |
| OOD | Stage 1 only | 0.313 | 0.200 |
| OOD | Full model (two-stage) | **0.945** | **0.021** |

diagonal affine transformation:

$$\tilde{\mathbf{x}}_t = \mathbf{s} \odot \mathbf{x}_t + \mathbf{b}, \tag{72}$$

where $\mathbf{s}, \mathbf{b} \in \mathbb{R}^p$ are learnable parameters trained by minimizing the CORAL loss (Sun et al., 2016) between source and target covariances. This was not required in any of our main experiments but may be useful when the mixing function $g$ induces distribution shifts not captured by mechanism interpolation alone.

### E.6. Detailed Limitations Discussion

We expand the four scope conditions summarised in Section 7.

**(i) Geometric capacity.** The bound $K \leq d+1$ is sufficient but not necessary; structurally sparse perturbations and few simultaneously active mechanisms relax it to $K_{\text{active}} \leq d+1$, allowing $K_{\text{total}}$ to grow well beyond $d$ (Section 6.7, Ap-

pendix E.11). A continuous parameterization $W(t) = h(c_t)$ removes the discrete-mixture enumeration entirely at the cost of explicit interpretability.

**(ii) Pure-domain training.** The framework assumes labeled pure-regime data, but degrades gracefully under both label noise and mechanism contamination (Weight Corr stays $\geq 0.93$ at $50\%$ contamination; Appendix E.7), with $\sigma_{\min}$ acting as a pre-deployment diagnostic. Unsupervised domain discovery (e.g., via learned soft gating with post-hoc clustering) is a natural future direction.

**(iii) Knowing $K$.** Mild over-specification incurs only a small cost (0.966 vs. 0.971), and duplicate experts can be detected via $\sigma_{\min}$; under-specification is recoverable for seen domains but not for absent ones (Appendix E.8).

**(iv) Real-world evaluation.** The vehicle and gait experiments rely on physically motivated proxies (instantaneous direction for vehicles, hip joint speed for gait), and are therefore evaluated by Pearson correlation, which is invariant to monotonic warping. This sacrifices absolute-scale recovery but preserves trajectory shape, which is the quantity of interest for transition analysis.

### E.7. Robustness to Impure Training Domains

We probe TRACE under two failure modes for the pure-domain assumption: (i) noisy domain labels at training time, and (ii) impurity at the data-generating-process level, where each domain's transition matrix is contaminated by other domains' perturbations.

**Label noise.** We randomly flip a fraction $\varepsilon$ of training samples' domain labels to a uniformly chosen incorrect domain. The shared encoder's reconstruction objective is label-agnostic, and individual experts tolerate moderate contamination from mis-routed samples.

**Mechanism impurity.** We contaminate each domain's transition matrix as

$$W_k = W_{\text{base}} + (1 - \varepsilon)\,\delta W_k + \varepsilon \cdot \text{mean}\big(\delta W_{j \neq k}\big),$$

with all entries of $\delta W$ nonzero. Under this contamination model, each domain's specific signal vanishes at the critical threshold $\varepsilon^* = (K_{\text{total}} - 1)/K_{\text{total}}$; for $K_{\text{total}} = 5$ this gives $\varepsilon^* \approx 0.80$, so $\varepsilon = 0.50$ is well below the limit.

Degradation is gradual and matches Theorem 4.2: as $\sigma_{\min}$ shrinks monotonically from 0.168 to 0.049, the error bound $O(1/\sigma_{\min})$ predicts proportionally worse recovery, and the empirical Weight Corr tracks this prediction. TRACE is therefore robust to both label-level and mechanism-level impurity, and $\sigma_{\min}$ provides a practical diagnostic for domain separability before deployment.

*Table 15.* Robustness to mechanism-level impurity ($K_{\text{active}} = 3$, $K_{\text{total}} = 5$). $\sigma_{\min}$ of the basis matrix $\hat{B}$ acts as a pre-deployment diagnostic of domain separability.

| $\varepsilon$ | MCC ↑ | Weight Corr ↑ | $\sigma_{\min}$ |
|---|---|---|---|
| 0% | $0.984 \pm 0.016$ | $0.996 \pm 0.001$ | 0.168 |
| 20% | $0.975 \pm 0.002$ | $0.981 \pm 0.002$ | 0.126 |
| 50% | $0.943 \pm 0.032$ | $0.936 \pm 0.033$ | 0.049 |

### E.8. Misspecified Number of Mechanisms $K$

We test misspecification at training time relative to the true $K_{\text{total}} = 5$.

**Over-specification ($K_{\text{train}} = 7$).** We add two extra domains, either by duplicating an existing mechanism or by averaging $W$ matrices from existing domains, and use all seven as basis. When the test trajectory's active domains do not include duplicate experts, Weight Corr $= 0.966$, close to the $K_{\text{train}} = 5$ baseline (0.971). When active domains do include duplicates, $\boldsymbol{\alpha}$ recovery degrades to Weight Corr $= 0.553$ because the solver cannot distinguish near-identical basis vectors. However, the recovered transition dynamics $W(t)$ remain accurate, since duplicate experts share the same $W^{(k)}$: any $\boldsymbol{\alpha}$ split between them yields identical $W(t) = \sum_k \alpha_k(t) W^{(k)}$. Practitioners can detect and merge duplicates via the $\sigma_{\min}$-based analysis above or post-hoc clustering.

**Under-specification ($K_{\text{train}} = 3$).** With two missing domains, recovery for trajectories whose active domains are all seen during training remains strong (Weight Corr $= 0.949$). For trajectories involving unseen domains, neither $\boldsymbol{\alpha}$ nor $W(t)$ can be recovered (Weight Corr $= 0.244$), as their dynamics were never learned – this is an inherent information limit.

**Recommendation.** Mild over-specification combined with $\sigma_{\min}$-based duplicate detection is a safe default. Over-specification causes only minor degradation (0.966 vs. 0.971), and under-specification degrades gracefully for seen domains (0.949).

### E.9. Simplex Constraint Ablation

We compare simplex-constrained against unconstrained $\boldsymbol{\alpha}(t)$ recovery on out-of-distribution transition data (active domains $\{0, 2, 4\}$, $T = 50$ steps, 500 trajectories).

*Table 16.* Simplex vs. unconstrained recovery on OOD transitions.

| Mode | Weight Corr ↑ | % Neg. $\alpha$ | Range of $\alpha$ |
|---|---|---|---|
| Simplex | **0.956** | 0% | $[0.03,\ 0.86]$ |
| Unconstrained | 0.971 | 18.0% | $[-0.10,\ 1.01]$ |

The unconstrained mode achieves a slightly higher Weight Corr (**+0.015**), but $18\%$ of recovered time steps assign physically meaningless negative weights (e.g., "negative running" has no interpretation). The simplex constraint is motivated by physical interpretability rather than accuracy: $\alpha_k$ represents mechanism $k$'s contribution at time $t$. We thus propose the *fraction of invalid $\boldsymbol{\alpha}$* as a concrete interpretability metric. The accuracy cost of the simplex is negligible (0.956 vs. 0.971, both above 0.95), while it guarantees physically valid outputs.

### E.10. Sample Efficiency: Extended Table

The main-text Sample Efficiency analysis (Section 6.6, Table 4) summarises trajectory recovery as training data shrinks from $100\%$ to $1\%$. For reference, Table 17 lists the same results with explicit per-domain sample counts ($N$/domain).

### E.11. Scaling: Mathematical Explanation and Extension

This appendix gives the mathematical explanation behind the empirical results in Section 6.7 (Table 5), and discusses a continuous parameterization for the $K_{\text{active}} \gg d$ regime.

**Why the looser bound holds.** The original constraint requires the basis matrix $\hat{B} \in \mathbb{R}^{d \times (K-1)}$ (whose columns are the domain-mean differences $\hat{\boldsymbol{\mu}}^{(k)} - \hat{\boldsymbol{\mu}}^{(0)}$) to have full column rank, which fails when $K > d+1$. However, trajectory recovery solves $\hat{B}\boldsymbol{\alpha}(t) = \hat{\mathbf{z}}(t) - \hat{\boldsymbol{\mu}}^{(0)}$ where the true $\boldsymbol{\alpha}(t)$ has only $K_{\text{active}}$ nonzero entries; recovery depends on the conditioning of the relevant $K_{\text{active}}$-column submatrix of $\hat{B}$, not the full matrix. The simplex constraint ($\alpha_k \geq 0$, $\sum_k \alpha_k = 1$) naturally favors sparse solutions, enabling stable recovery in the underdetermined regime $K > d+1$. Empirically, the binding constraint is therefore $K_{\text{active}} \leq d+1$, not $K_{\text{total}} \leq d+1$.

**Continuous parameterization for $K_{\text{active}} \gg d$.** For scenarios where many mechanisms are simultaneously active, one can replace the discrete mixture $W(t) = \sum_k \alpha_k(t) W^{(k)}$ with a continuous parameterization $W(t) = h(c_t)$, where $c_t \in \mathbb{R}^{d_c}$ is a low-dimensional trajectory and $h$ maps it directly to mechanism space, bypassing discrete enumeration entirely. TRACE prioritizes interpretability via explicit mixing coefficients, but this extension would handle arbitrarily many mechanisms at the cost of that interpretability.

### E.12. Naive Heuristic Baselines on Real Data

To contextualize the difficulty of trajectory recovery on real data, we compare TRACE against domain-agnostic physical heuristics on the vehicle dataset (UAVDT).

The heuristics produce erratic sawtooth oscillations, while TRACE recovers smooth trajectories that align with the physical transition dynamics. This demonstrates that the gain over discrete-switching baselines (NCTRL, Section 6.4.1) is not simply due to the heuristic ease of the dataset.

### E.13. Connection to Dynamic Bayesian Networks

Each $W^{(k)}$ in our formulation plays the role of a lagged-causal-effect (transition) matrix in a Dynamic Bayesian Network (DBN), defining $p(\mathbf{z}_t \mid \mathbf{z}_{t-1}, k)$ for atomic mechanism $k$. The mixing trajectory $\boldsymbol{\alpha}(t)$ then parameterizes a time-varying DBN

$$W(t) = \sum_{k=0}^{K-1} \alpha_k(t) \, W^{(k)},$$

generalizing window causal graphs (which assume fixed causal structure within each discrete time interval) to continuously varying transition weights. We do not impose DAG constraints on $W^{(k)}$ because our model captures only inter-temporal (lagged) effects with no instantaneous edges, which makes the unrolled full-time graph inherently acyclic for any lag order $L$. Equivalently, latent edges connect $\mathbf{z}_{t-L:t-1} \to \mathbf{z}_t$ across time, so the time-unrolled DAG-ness is preserved by construction.

### E.14. Future Work and Direct Applications

This appendix elaborates on the future-work directions sketched in Section 7 and lists deployment settings where TRACE is directly applicable.

**Algorithmic extensions.** Three near-term extensions follow directly from our analysis.

**(1) Automatic $K$ inference.** Mild over-specification combined with $\sigma_{\min}$-based duplicate detection (Appendix E.8) already provides a practical recipe. A principled next step is a model-selection criterion based on the conditioning of the basis matrix $\hat{B}$: increase $K_{\text{train}}$ until newly added basis columns become near-collinear with existing ones (detected by a sharp drop in $\sigma_{\min}$). This avoids manual tuning and makes the framework deployment-ready in settings where the true mechanism count is unknown, connecting to broader efforts on principled identification under partial observability such as nonparametric inference of counterfactual distributions under confounding (Sun & Zhang, 2026).

**(2) Unsupervised mechanism discovery.** The pure-regime requirement can be relaxed by replacing the deterministic one-hot gating with a learned soft gating network—for instance, a multi-head attention module whose theoretical advantages over single-head gating are now well-characterized (Cui et al., 2025)—that clusters unlabeled

*Table 17.* Sample efficiency with explicit per-domain sample counts. Same numbers as Table 4; included for reproducibility.

| Fraction | N/domain | MCC | Weight Corr ($K = 3$) | Weight Corr ($K = 5$) |
|---|---|---|---|---|
| 100% | 40,000 | 0.963 | 0.986 | 0.979 |
| 20% | 8,000 | $0.886 \pm 0.088$ | $0.968 \pm 0.014$ | $0.955 \pm 0.022$ |
| 10% | 4,000 | $0.752 \pm 0.163$ | $0.960 \pm 0.013$ | $0.941 \pm 0.014$ |
| 1% | 400 | $0.622 \pm 0.069$ | $0.688 \pm 0.377$ | $0.640 \pm 0.169$ |

*Table 18.* TRACE vs. domain-agnostic heuristics on the vehicle turning dataset.

| Method | Weight Corr ↑ |
|---|---|
| TRACE | **0.960** |
| Centroid tracking | 0.911 |
| Optical flow | 0.902 |

data into mechanism groups. Once latent variables are identifiable under Theorem 4.1, the domain structure may be recoverable via clustering in latent space, drawing on tools from graph-based multi-manifold clustering (Trillos et al., 2023). Our impurity experiments (Appendix E.7) show that latent separability is preserved even at 50% contamination, suggesting the latent space remains usable for downstream domain discovery.

**(3) Nonparametric mechanism compositions.** The discrete-mixture form $W(t) = \sum_k \alpha_k(t)W^{(k)}$ can be generalized to a continuous parameterization $W(t) = h(c_t)$ with $c_t \in \mathbb{R}^{d_c}$ a low-dimensional latent trajectory (Appendix E.11). This handles arbitrarily many mechanisms at the cost of explicit mixing coefficients, and is the natural choice when interpretability is secondary to coverage. This direction parallels recent work on latent-state time-series forecasting (Yang et al., 2026a), distribution-aware temporal alignment (Hu et al., 2026), information-bottleneck-based imputation under partial observability (Yang et al., 2026b), and active intervention on latent manifolds (Xu et al., 2026).

**Direct application settings.** TRACE is most readily deployable in domains where individual regimes are easy to instrument but transitions are scarce. Four immediate examples:

**(a) Manufacturing operating modes.** A production line typically operates in a finite set of distinct configurations (different products, speeds, or material grades), with transitions occurring during retooling or shift changes. Pure-regime telemetry is abundant from steady-state operation, while transition data is rare. TRACE recovers smooth interpolations between operating modes that classical change-point methods would force into discrete states.

**(b) Clinical disease staging.** Population studies routinely collect cross-sectional patient data tagged with discrete disease stages, but longitudinal recordings of smooth disease progression are expensive and ethically constrained. TRACE can be trained on stage-labeled snapshots and then track an individual patient's continuous progression at inference time, enabling early-warning applications. Such progression-aware modeling complements recent advances in causal medical image classification (Liu et al., 2024) and hierarchical copula-based survival analysis with competing risks (Liu et al., 2025).

**(c) Robotic locomotion and contact dynamics.** Free-space dynamics and contact-rich regimes can each be recorded in isolation, but smooth transitions (e.g., from free swing to ground contact) are short and difficult to capture exhaustively. TRACE recovers the time-varying interpolation between these regimes, which is the quantity needed for predictive control during transitions. Related regime-transition dynamics also arise in sensor-based human activity recognition, where diffusion mappings between wearable signals and skeletal motion span multiple movement modes (Sharma et al., 2025).

**(d) Sequential modeling and multimodal interaction.** The framework extends naturally to time-series settings where regimes are interpretable but smoothly evolving, including continuous-time sequential recommendation (Fan et al., 2021), long-context video-LLM frameworks that span multiple temporal scales (Shang et al., 2024), tool-integrated multimodal agents whose operating mode shifts continuously across tasks (Lei et al., 2024; 2026), and cross-modal generation pipelines where conditional regimes evolve smoothly along the input (e.g., dance-to-music synthesis (Sun et al., 2025a)).

In each setting, TRACE recovers continuous mixing coefficients without requiring labeled transition trajectories—directly addressing the data scarcity that makes discrete-switching methods impractical.

