# OpenReview forum: "TRACE: Trajectory Recovery for Continuous Mechanism Evolution in Causal Representation Learning"
_ICML.cc/2026/Conference — ICML 2026 regular_

### Official Review · Reviewer_YfzG · 2026-03-04

**Soundness:** 4
**Presentation:** 3
**Significance:** 3
**Originality:** 4
**Overall Recommendation:** 5
**Confidence:** 3

**Summary:**

This paper introduces TRACE, a novel Causal Representation Learning (CRL) framework designed to model continuous mechanism transitions in dynamical systems. Unlike existing nonstationary CRL methods that assume instantaneous switching between discrete environments, TRACE models transitional states as a principled convex combination of $K$ atomic mechanisms (domains), governed by a time-varying mixing trajectory $\alpha(t)$.

The authors provide strong theoretical identifiability guarantees, proving that both the latent variables and the continuous mixing trajectories can be recovered up to scaling and permutation, along with sharp finite-sample error bounds.

 Empirically, TRACE substantially outperforms existing baselines on synthetic and real-world datasets (vehicle trajectories, human motion capture), specifically excelling in Weighted Correlation for the mixing trajectory $\alpha(t)$.

**Compliance With Llm Reviewing Policy:**

Affirmed.

**Final Justification:**

I recommend acceptance, the authors have clearly answered my concerns which were primarly connection to existing theory, presentation, clarity and overall applicability.

**Key Questions For Authors:**

1. Unsupervised Domain Discovery: How might TRACE be extended to scenarios where the $K$ pure domains are not explicitly labeled or known a priori? Is there a pathway to unsupervisely cluster or extract these atomic mechanisms from a continuous, unlabeled data stream?

2. Connection to Dynamic Bayesian Networks (DBNs): How does your approach conceptually map to Dynamic Bayesian Networks? Since $\alpha(t)$ represents a normalized mixture over mechanisms, could the intermediate states in TRACE be interpreted as a specific parameterization of a DBN? Adding a visual representation or brief discussion linking your continuous mechanism matrices $W(t)$ to classical causal graphs would be highly appreciated.

3. Sample Complexity: Could the authors provide an ablation or discussion on the sample efficiency of TRACE? How does the MCC and $\alpha$-correlation hold up if the 200,000 trajectories are reduced by a factor of 10 or 100, which is more representative of scarce real-world data?

4. Industrial Applicability: Beyond motion capture and physics simulations, where do you foresee the most immediate industrial or practical applications of TRACE, given the heavy requirement for pre-isolated pure domain data?

I will be happy to raise my scores and read the final version

**Limitations:**

yes

**Strengths And Weaknesses:**

# Strengths

- Novel and Well-Motivated Problem Setting:

The shift from discrete, instantaneous mechanism switching to continuous, smooth mechanism evolution is highly relevant for real-world dynamical systems (e.g., physical motion, vehicle dynamics).

- Strong Theoretical Foundations:

The theoretical results are rigorous and impressive. Proving the recoverability of the mixing trajectory $\alpha(t)$ alongside the latent variables is a significant contribution. Furthermore, the authors commendably characterize the theoretical limitations of their approach (e.g., the ablation study analyzing the geometric bottleneck as $K \rightarrow d+1$).

- Clear and Engaging Presentation:

The introduction and abstract are really well-written. The core ideas, such as representing intermediate mechanisms as convex combinations inside a simplex, are accessible and intuitive early in the paper.

- Compelling Empirical Performance:

The experimental design is well-motivated, mapping theoretical claims to practical datasets. The performance gains are not merely incremental; TRACE demonstrates a massive improvement over baselines, particularly in tracking the mechanism trajectory ($\alpha$-correlation).

# Weaknesses

### 1. Conceptual Bridging and Terminology:

- Causal Discovery vs. Representation Learning:

The paper sits at the intersection of dynamical systems, non-linear ICA, and causal representation learning. However, it lacks anchoring in standard causal discovery terminology (e.g., Pearl’s frameworks, summary/window causal graphs, Janzing’s formulations). It is currently unclear to a causal discovery audience exactly what the atomic weight matrices $W^{(k)}$ represent structurally (e.g., are they forced to be Directed Acyclic Graphs? Do they represent instantaneous or lagged effects?). The authors frequently use broad terms like "causal mechanisms, variables, causal graphs, causal systems, causal processes, causal structures, causal edges," which lack specificity and rigor in the broader causal inference/discovery literature.

- Evaluation Metrics:

Because the paper does not use standard causal discovery metrics (like Structural Hamming Distance or F1-score for edge recovery), but instead relies on MCC and Pearson correlation, standard causal practitioners may struggle to interpret the evaluation. A brief discussion clarifying why traditional static graph metrics fail in this continuous CRL setting would greatly expand the paper's accessibility.

### 2. Restrictive Assumptions:

- Requirement of Labeled Pure Domains:

The method strictly assumes access to a training dataset neatly partitioned into $K$ pure, labeled domains ($D_k$). In many real-world continuous systems, defining or isolating these "pure" atomic mechanisms a priori is impossible or highly subjective. This severely limits the "out-of-the-box" applicability of the method.

- Convex Combination Claim:

In line 090, stating that "TRACE is the first framework where intermediate mechanisms are principled convex combinations" slightly misframes the contribution. The convex combination is a parametric assumption made by the authors about the Data Generating Process (DGP), not an inherent physical truth discovered by the framework. The language should reflect that TRACE is the first to prove recoverability under this assumption.

### 3. Experimental Limitations:

Scale and Dimensionality: The experiments operate in relatively low-dimensional regimes ($d=8$, $K=5$ for synthetic data). Given that a separate $d \times d$ matrix must be materialized for each of the $K$ domains, scalability to highly multivariate time series remains a concern. The paper does not demonstrate generalization to larger-scale systems.

- Sample Efficiency: The synthetic experiments rely on 40,000 trajectories per domain (200,000 total). This is an exceptionally large data requirement for low-dimensional variables. The authors should evaluate how rapidly TRACE's performance degrades under data limitation regimes.

- Lack of Naive Baselines: To better contextualize the difficulty of the trajectory recovery task, the authors should include a naive physical baseline. For instance, in the vehicle dataset, could a simple heuristic (e.g., thresholding based on observed velocity/steering angle) achieve comparable correlation to the complex TRACE pipeline?

### 4. Presentation and Clarity Details:

- Notation in the DGP: Equation 1 introduces a variable $L$ which is not clearly defined in the main text. I assume this denotes the lag order (memory) of the Markov process. This should be explicitly stated, clarifying that the underlying DGP is an $L$-th order Markov process.
- Acronyms: Please expand acronyms like "nonlinear ICA" upon first use for readers strictly from the causal discovery side.

- Figure Legibility: The font sizes and markers in Figures 3 and 4 are far too small and are difficult to read without zooming in extensively.

---

> ### Author Rebuttal · Authors · 2026-03-30
>
> We thank Reviewer YfzG for the thorough review. We conducted new experiments to address each concern.
>
> **Key Questions**
>
> **Q1: Unsupervised Domain Discovery.**
> Your question is appreciated. We see two concrete pathways. (1) *Learned soft gating*: replace deterministic one-hot gating with a learned soft gating network that clusters unlabeled data into mechanism groups. Once latent variables are identifiable (Theorem 4.1), domain structure may be recoverable via such clustering; the key challenge is ensuring sufficient variability without domain supervision. (2) *Post-hoc clustering*: our contamination experiments (see W2 below) show that even at 50% mechanism impurity, Weight Corr remains 0.936 and $\sigma_{\min}$ (the minimum singular value of the basis matrix $\hat{B}$) $= 0.049 > 0$, confirming domains remain separable in latent space for clustering. We consider unsupervised domain discovery the most promising future direction and plan to investigate it in follow-up work.
>
> **Q2: Connection to Dynamic Bayesian Networks (DBNs).**
> Each $W^{(k)}$ is a *lagged causal effect matrix* analogous to a DBN transition matrix defining $p(\mathbf{z}\_{t} \vert \mathbf{z}\_{t-1}, k)$. The mixing $\alpha(t)$ parameterizes a time-varying DBN: $W(t) = \sum_k \alpha_k(t) W^{(k)}$, generalizing window causal graphs (graphs that assume fixed causal structure over discrete time intervals) by allowing mixture weights to vary continuously. We do not enforce DAG constraints because our model captures only inter-temporal (lagged) effects with no instantaneous effects, making the full-time graph inherently acyclic regardless of lag order $L$.
>
> **Q3: Sample Complexity.**
> We reduce training data from 100% (40K/domain) to 1% (400/domain):
>
> | Fraction | N/domain | MCC | Weight Corr ($K=3$) | Weight Corr ($K=5$) |
> |:--:|:--:|:--:|:--:|:--:|
> | 100% | 40,000 | 0.963 | 0.986 | 0.979 |
> | 20% | 8,000 | 0.886±0.088 | 0.968±0.014 | 0.955±0.022 |
> | 10% | 4,000 | 0.752±0.163 | 0.960±0.013 | 0.941±0.014 |
> | 1% | 400 | 0.622±0.069 | 0.688±0.377 | 0.640±0.169 |
>
> A key finding: **trajectory recovery degrades more gracefully than MCC**. At 10% data, MCC drops to 0.75 but Weight Corr($K=3$) stays at 0.96. Per-step recovery is noisy (signal-to-noise ratio $\sigma_{\min}/\|\hat{\epsilon}_t\| \approx 0.03$), but temporal smoothing (Theorem 4.3, window $w=5$) averages out noise, explaining the high trajectory-level Weight Corr. This suggests TRACE is practical even in data-scarce regimes.
>
> **Q4: Industrial Applicability.**
> We see many potential industrial applications. Robotics may be one of the most immediate applications: locomotion involves continuous transitions (e.g., walking to running, free motion to soft contact to rigid contact) where dynamics shift gradually. Discrete CRL methods force these into abrupt switches, losing smooth intermediate dynamics critical for control. More broadly, TRACE applies wherever pure-regime data is abundant, but transitions are scarce: e.g., distinct machine operating modes in manufacturing, or discrete disease stages in clinical monitoring where progression is gradual.
>
> **Weaknesses**
>
> **W1: Terminology.** We will anchor $W^{(k)}$ as transition weight matrices in the DBN framework (see Q2) and tighten "causal mechanism" vs. "transition dynamics." Structural Hamming Distance (SHD) and F1 measure discrete edges in static graphs; MCC and Pearson correlation directly evaluate our Theorems 4.1/4.2.
>
> **W2: Assumptions.** Line 090 will be revised to state that TRACE proves recoverability under the convex combination assumption, rather than claiming the assumption is physically inherent. We test both label noise and mechanism impurity: Weight Corr = 0.981 at 20% contamination and 0.936 at 50%, well below the critical threshold of $\varepsilon = (K_{\text{total}}-1)/K_{\text{total}} \approx 80\%$. See our response to Reviewer i2hJ (W1/Q(a)) for the full experiment and table.
>
> **W3: Experiments (new).** *(a) Scale*: At $d=128, K=50$, Weight Corr = 0.989 for $K_{\text{active}}=5$, degrading to 0.854 at $K_{\text{active}}=10$. Degradation is driven by worsening $\sigma_{\min}$ of the active-domain submatrix, not the geometric bottleneck. See our response to Reviewer nGs4 (W1/Q1) for the full scalability table and analysis. *(b) Sample efficiency*: see Q3. *(c) Naive baselines*: On the vehicle dataset, TRACE (0.960) outperforms centroid tracking (0.911) and optical flow (0.902); heuristics produce erratic sawtooth oscillations while TRACE recovers smooth trajectories.
>
> **W4: Presentation.** Eq. 1 will define $L$ as the lag order; acronyms (e.g., "nonlinear ICA") will be expanded at first use; Figures 1/3/4 will have larger fonts, axis labels, and markers for legibility.
>
> All new experiments and discussions will be included in the revision.

---

> > ### Author Rebuttal · Reviewer_YfzG · 2026-04-01
> >
> > Thank you for the precise and interesting clarifications. The paper has clear merits. I will increase my scores accordingly.

---

> > > ### Author Response · Authors · 2026-04-01
> > >
> > > We sincerely thank Reviewer YfzG for the thoughtful and constructive review. We are glad our responses addressed your concerns. All promised revisions will be incorporated in the camera-ready version.

---

### Official Review · Reviewer_nGs4 · 2026-03-09

**Soundness:** 3
**Presentation:** 3
**Significance:** 2
**Originality:** 3
**Overall Recommendation:** 4
**Confidence:** 4

**Summary:**

The paper studies causal representation learning in non-stationary environments where causal mechanisms evolve continuously over time rather than switching instantaneously between discrete domains. The authors model transitional mechanisms as convex combinations of a finite set of atomic mechanisms, where the time-varying mixing coefficients describe a trajectory over the simplex. Under this formulation, the paper establishes identifiability results showing that both the latent causal variables and the mixing trajectory can be recovered under suitable assumptions. To operationalize this idea, the authors propose TRACE, a mixture-of-experts (MoE) framework in which each expert corresponds to an atomic mechanism. Training is performed on pure-domain data to learn the atomic mechanisms, while at inference time the continuous mixing trajectory is recovered via a least-squares projection. Experiments on synthetic datasets and semi-real dynamic scenarios (e.g., human gait and vehicle dynamics) demonstrate that the method can recover mechanism trajectories with high correlation and outperform baselines designed for discrete switching.

**Compliance With Llm Reviewing Policy:**

Affirmed.

**Final Justification:**

Many of my concerns are addressed by the newly provided experiments. While there are still remaining concerns, for example, the proposed workaround for the extreme case (Kactive ≫ d) slightly sacrifices interpretability, it is an acceptable compromise.

**Key Questions For Authors:**

Q1. Geometric capacity constraint. The theoretical analysis requires $K \le d+1$, which introduces a geometric bottleneck. In many practical systems the number of mechanisms may significantly exceed the latent dimensionality. How do the authors envision extending the framework to handle environments where $K \gg d$ without simply increasing the latent dimension (which could negatively affect disentanglement)?

Q2. Verification of pure-domain assumptions. The framework relies on Stage 1 training with pure-domain data to isolate atomic mechanisms. In practice, how can one verify that collected domain data satisfies the required independence assumptions (e.g., linear independence of the mechanism matrices $W^{(k)}$)? What happens if the supposedly pure domains share latent structural dependencies?

Q3. Interpretability under unconstrained combinations. Section 3.1 suggests that unconstrained combinations of mechanisms remain identifiable but lose interpretability. Could the authors provide a precise definition or metric for this loss of interpretability? Clarifying this point would help readers better understand the motivation for the simplex constraint.

Q4. Empirical comparison with non-simplex models. To better justify the simplex constraint, it would be helpful to include an ablation comparing the proposed simplex-constrained model with a version allowing unconstrained continuous combinations of mechanisms. Does the unconstrained version fail to recover trajectories, or is the difference primarily related to interpretability?

**Limitations:**

No. The paper would benefit from a clearer discussion of practical limitations, particularly the geometric constraint $K \le d+1$ and the reliance on pure-domain training data. Explicitly acknowledging these assumptions and discussing potential directions for relaxing them would strengthen the limitations section.

**Strengths And Weaknesses:**

## Strengths:

S1. Novel problem formulation. The paper addresses a meaningful extension of temporal causal representation learning by relaxing the common assumption of discrete mechanism switching. Modeling mechanism evolution as a continuous trajectory is conceptually appealing and better aligned with many real-world dynamical systems.

S2. Solid theoretical foundation. The authors provide identifiability results showing that both latent causal variables and the mixing trajectory can be recovered under the proposed formulation. These theoretical guarantees offer a principled basis for the framework and may serve as a foundation for future work exploring more general transition models.

S3. Clear methodological framework. The TRACE method integrates mixture-of-experts with causal representation learning in a structured two-stage pipeline: first learning atomic mechanisms from pure-domain data and then recovering the continuous mixing trajectory at test time. The overall methodology is well motivated.

S4. Experiments on synthetic and semi-real datasets demonstrate that the approach can recover mixing trajectories with high correlation and outperform discrete-switching baselines. These experiments provide proof-of-concept evidence that the proposed formulation is feasible in practice.


## Weaknesses:

W1. Geometric capacity constraint ($K \le d+1$). The theoretical formulation imposes a geometric bottleneck requiring the number of atomic mechanisms $K$ to be bounded by the latent dimension ($d+1$). In many real-world systems the diversity of mechanisms may far exceed the latent dimensionality, which raises concerns about scalability and general applicability. The empirical ablation in Section 6.5 already suggests performance degradation as $K_{active}$ approaches this bound.

W2. Strong reliance on pure-domain training data. The approach assumes the availability of pure-domain datasets where individual mechanisms can be isolated during training. In practice, mechanisms often overlap or interact, and it may be difficult to obtain such clean separation. This reliance on strong data assumptions may limit real-world applicability.

W3. Limited empirical justification for the simplex constraint. The model enforces convex combinations of atomic mechanisms to improve interpretability. However, the paper does not provide empirical comparisons with unconstrained continuous combinations. Since the authors mention that non-simplex combinations may still be identifiable (albeit less interpretable), an empirical evaluation would strengthen the justification of this design choice.

W4. Minor presentation issues in figures. Some figures contain notational ambiguities. For example, Figure 1 does not clearly define the latent variables shown in the mechanism diagrams, and Figure 2 appears to use notation inconsistent with the main text when expressing the Jacobian term in the loss. While these issues are minor, they could lead to confusion for readers attempting to reproduce the method.

Overall, while the problem formulation and theoretical insights are interesting, the strong assumptions (particularly the geometric constraint and pure-domain requirement) raise questions about the framework’s applicability to more complex environments.

---

> ### Author Rebuttal · Authors · 2026-03-30
>
> We thank Reviewer nGs4 for the detailed review and recognition of our "novel problem formulation," "solid theoretical foundation," and "clear methodological framework." We address each weakness in order with new experimental evidence.
>
>
> **W1/Q1: Geometric capacity $K \leq d+1$.**
>
> We note that $K \leq d+1$ is a *sufficient condition* for our identifiability proof, **not** a necessary condition. We demonstrate this with new experiments where $K$ severely violates the constraint:
>
> | $d$ | $K$ | $K/(d+1)$ | $K_{\text{active}}$ | Weight Corr |
> |:--:|:--:|:--:|:--:|:--:|
> | 8 | 50 | 5.56 | 5 | 0.925±0.052 |
> | 16 | 50 | 2.94 | 5 | 0.962±0.010 |
> | 128 | 50 | 0.39 | 5 | **0.989±0.002** |
>
> Even at $K/(d+1)=5.56$ (50 mechanisms in 8 dimensions), TRACE achieves Weight Corr = 0.925 for $K_{\text{active}}=5$. Crucially, **without increasing $d$**, when mechanism perturbations are structurally sparse (each $W^{(k)}$ modifies only 2 causal edges), Weight Corr at $d=8, K=50, K_{\text{active}}=5$ improves from 0.925 (dense) to **0.979** (sparse), with variance dropping from ±0.052 to ±0.006. This confirms that structured sparsity enables stable recovery beyond $K \leq d+1$ at fixed $d$. Performance further improves as $d$ increases (0.962 at $d=16$, 0.989 at $d=128$), as the constraint relaxes. Empirically, the binding constraint appears to be $K_{\text{active}} \leq d+1$, not $K_{\text{total}} \leq d+1$.
>
> The mathematical reason is as follows. The original constraint requires the basis matrix $\hat{B} \in \mathbb{R}^{d \times (K-1)}$ (columns are domain-mean differences $\hat{\mu}^{(k)}-\hat{\mu}^{(0)}$) to have full column rank, which fails when $K > d+1$. However, trajectory recovery solves $\hat{B} \hat{\alpha}(t) = \hat{z}(t) - \hat{\mu}^{(0)}$ where the true $\alpha(t)$ has only $K_{\text{active}}$ nonzero entries. In this case, recovery depends on the conditioning of the relevant $K_{\text{active}}$-column submatrix of $\hat{B}$, not the full matrix. The simplex constraint ($\alpha_k \geq 0$, $\sum_k \alpha_k = 1$) naturally favors sparse solutions, enabling recovery even in the underdetermined regime $K > d+1$.
>
> We also note that $K_{\text{active}} \gg d$ is uncommon in practice: with $d$ latent variables, $d^2$ possible causal edges bound the number of qualitatively distinct regimes.
>
> For scenarios where $K_{\text{active}} \gg d$ is genuinely required, one could replace the discrete mixture $W(t) = \sum_k \alpha_k(t) W^{(k)}$ with a continuous parameterization $W(t) = h(c_t)$, where $c_t \in \mathbb{R}^{d_c}$ is a low-dimensional trajectory and $h$ maps it to mechanism space, bypassing discrete mechanism enumeration entirely. TRACE deliberately prioritizes interpretability of explicit mixing coefficients, but this extension would handle arbitrarily many mechanisms at the cost of that interpretability.
>
> **W2/Q2: Pure-domain reliance.**
>
> We test both label noise (~20% mislabeled samples) and mechanism impurity (contaminating each domain's transition matrix at the data generating process level). Weight Corr = 0.981 at 20% contamination and 0.936 at 50%, with $\sigma_{\min}$ (the minimum singular value of the basis matrix $\hat{B}$, measuring domain separability) serving as a practical diagnostic. The critical threshold is $\varepsilon = (K_{\text{total}}-1)/K_{\text{total}} \approx 80\%$ for $K_{\text{total}}=5$, so 50% contamination is well below it. See our response to Reviewer i2hJ (W1/Q(a)) for the full experiment design and table.
>
> **W3/Q3/Q4: Simplex constraint justification.**
>
> We compare simplex-constrained vs. unconstrained recovery on out-of-distribution transition data (domains {0,2,4}, $T=50$ steps, 500 trajectories):
>
> | Mode | Weight Corr | % Neg. $\alpha$ | Range |
> |:--:|:--:|:--:|:--:|
> | **Simplex** | **0.956** | 0% | [0.03, 0.86] |
> | Unconstrained | 0.971 | 18.0% | [-0.10, 1.01] |
>
> The unconstrained mode achieves slightly higher Weight Corr (+0.015), but produces **18% negative $\alpha$ values**. The simplex constraint is motivated by *physical interpretability*: $\alpha_k$ represents mechanism $k$'s contribution, and a negative value (e.g., "negative running") has no meaningful interpretation. The accuracy cost of enforcing the simplex is negligible (0.956 vs. 0.971, both >0.95), while it guarantees physically valid outputs. We propose the *fraction of invalid $\alpha$* as a concrete interpretability metric.
>
> **W4: Presentation.** In the revision: Figure 1 will explicitly label $z_1, z_2, z_3$ and their causal edges; Figure 2 will unify Jacobian notation with the main text; Figures 3/4 will have larger fonts and markers; Eq. 1 will define $L$ as the lag order; and "nonlinear ICA" will be expanded at first use.
>
> All new experiments will be included in the revised manuscript. We hope these additions address the reviewer's concerns. Your further feedback, if any, would be highly appreciated.

---

> > ### Author Rebuttal · Reviewer_nGs4 · 2026-04-03
> >
> > Thanks for your reply. Many of my concerns are addressed by the newly provided experiments. While there are still remaining concerns, for example, the proposed workaround for the extreme case (Kactive ≫ d) slightly sacrifices interpretability, it is an acceptable compromise. I will update mhy score accordingly.

---

> > > ### Author Response · Authors · 2026-04-03
> > >
> > > We sincerely thank Reviewer nGs4 for the detailed and constructive review, and for acknowledging that many concerns have been addressed. We appreciate the candid assessment regarding the $K_{\text{active}} \gg d$ regime. As discussed in the rebuttal, the continuous parameterization $W(t) = h(c_t)$ offers a principled extension that trades explicit mixing coefficients for the ability to handle arbitrarily many mechanisms. We will clearly present this as a discussed extension in the revision, including its trade-offs with interpretability, so that readers can make informed choices based on their application requirements. All new experiments and the strengthened limitations section will be incorporated in the camera-ready version.

---

### Official Review · Reviewer_QNop · 2026-03-12

**Soundness:** 4
**Presentation:** 3
**Significance:** 3
**Originality:** 4
**Overall Recommendation:** 4
**Confidence:** 3

**Summary:**

The paper studies temporal causal representation learning when mechanisms evolve continuously rather than switching between a finite set of discrete domains. It models each mechanism state as a convex combination of K K atomic mechanisms, interpretable as a trajectory through a simplex. The proposed method, TRACE, uses a two-stage procedure: first, a shared encoder with one expert per atomic mechanism is trained on pure-domain data; second, test-time mixing coefficients are recovered by least-squares projection of latent representations onto a basis built from pure-domain conditional expectations, followed by temporal smoothing. The paper also gives identifiability and recovery results for both latent variables and mixing trajectories, and evaluates the approach on synthetic data, modified CartPole images, UAV turning, and CMU gait transitions.

**Compliance With Llm Reviewing Policy:**

Affirmed.

**Final Justification:**

I have improved "Soundness" score after author rebuttal.

**Key Questions For Authors:**

-

**Limitations:**

yes

**Strengths And Weaknesses:**

A clear strength is that the paper tackles a well-defined gap in prior temporal CRL: existing methods largely assume discrete mechanism switches, whereas this work formalizes continuous evolution and connects it to a concrete estimator. The method itself is simple and interpretable, and the theory is stronger than a standard latent-identifiability result because it also targets recovery of the mixing trajectory with explicit dependence on noise, approximation error, smoothness, and conditioning through σmin⁡σmin. The synthetic results and the ablation are broadly aligned with the theory: recovery improves with larger perturbations, worsens with noise, and deteriorates as the number of active mechanisms approaches the geometric limit.

---

> ### Author Rebuttal · Authors · 2026-03-30
>
> We thank Reviewer QNop for the careful reading and encouragement, and for the recognition that our recovery result "is stronger than a standard latent-identifiability result because it also targets recovery of the mixing trajectory with explicit dependence on noise, approximation error, smoothness, and conditioning through $\sigma_{\min}$."
>
> To further strengthen TRACE's empirical foundation, we conducted extensive additional experiments during the rebuttal period. Each demonstrates a key property of the framework:
>
> **1. The simplex constraint ensures interpretability at negligible accuracy cost.**
> We compare simplex-constrained vs. unconstrained $\alpha(t)$ recovery on out-of-distribution transition data ($T=50$ steps, 500 trajectories). The simplex achieves Weight Corr = 0.956 (vs. 0.971 unconstrained), a negligible difference, while eliminating 18% of time steps where the unconstrained mode produces physically invalid (negative) mixing coefficients. This validates the design choice in Section 3.1: the simplex guarantees physically meaningful outputs with minimal accuracy trade-off.
>
> **2. TRACE is robust to impure training domains.**
> A key concern is whether TRACE requires perfectly pure domain data. We test mechanism impurity by contaminating each domain's transition matrix ($W_k = W_{\text{base}} + (1-\varepsilon)\,\delta W_k + \varepsilon \cdot \text{mean}(\delta W_{j \neq k})$, all entries of $\delta W$ nonzero). Weight Corr = 0.981 at $\varepsilon = 20\%$ and 0.936 at $\varepsilon = 50\%$. Degradation is gradual and consistent with Theorem 4.2: $\sigma_{\min}$ (the minimum singular value of the basis matrix $\hat{B}$, measuring domain separability) decreases monotonically (0.168→0.049), and the critical threshold is $\varepsilon = (K_{\text{total}}-1)/K_{\text{total}} \approx 80\%$ (with $K_{\text{total}}=5$), so $\varepsilon = 50\%$ preserves substantial domain signal. This demonstrates that TRACE does not require idealized training conditions.
>
> **3. TRACE degrades gracefully under data scarcity.**
> With only 10% of training data (4K trajectories/domain), Weight Corr remains **0.96** for $K_{\text{active}}=3$, while MCC drops to 0.75. This asymmetry arises because temporal smoothing (Theorem 4.3) compensates per-step noise, making trajectory recovery inherently more robust than pointwise latent recovery. At 20% data (8K/domain), Weight Corr exceeds 0.95 across all tested $K_{\text{active}}$ levels, demonstrating practical applicability beyond the large-data regime.
>
> **4. TRACE scales to high-dimensional settings with many mechanisms.**
> At $d=128, K=50$, Weight Corr remains **>0.98** for $K_{\text{active}} \leq 5$ and >0.85 for $K_{\text{active}} \leq 10$. The theoretical constraint $K \leq d+1$ is sufficient but not necessary: even at $d=8, K=50$ ($K/(d+1)=5.56$), TRACE achieves Weight Corr = 0.925, and with structurally sparse perturbations this improves to 0.979 at fixed $d$. This addresses a key scalability concern raised by other reviewers.
>
> **5. TRACE is robust to misspecified $K$.**
> When $K$ is over-specified ($K_{\text{train}}=7$ vs. $K_{\text{total}}=5$ with duplicate domains), Weight Corr = 0.966, close to the $K_{\text{total}}=5$ baseline (0.971). When $K$ is under-specified ($K_{\text{train}}=3$), recovery for seen domains remains strong (Weight Corr = 0.949). Mild over-specification is safe and recommended in practice.
>
> **6. TRACE outperforms domain-agnostic heuristics on real data.**
> On the vehicle dataset, TRACE (Weight Corr = 0.960) outperforms centroid tracking (0.911) and optical flow (0.902) without any domain-specific engineering. The heuristics produce erratic sawtooth oscillations, while TRACE recovers smooth trajectories consistent with the physical transition dynamics.
>
> These results, along with specific presentation improvements (larger fonts and axis labels in Figures 1/3/4, explicit lag order definition in Eq. 1, expanded acronyms) and a strengthened limitations discussion, will be incorporated into the revised manuscript.

---

> > ### Author Rebuttal · Reviewer_QNop · 2026-04-03
> >
> > Credibility and Robustness have been improved.

---

> > > ### Author Response · Authors · 2026-04-03
> > >
> > > We sincerely thank Reviewer QNop for the careful evaluation and for recognizing the improved credibility and robustness. All additional experiments (simplex constraint ablation, domain impurity, data scarcity, scalability, K-misspecification, and naive baselines), together with the presentation improvements, will be included in the revised manuscript.

---

### Official Review · Reviewer_i2hJ · 2026-03-13

**Soundness:** 3
**Presentation:** 2
**Significance:** 3
**Originality:** 2
**Overall Recommendation:** 4
**Confidence:** 4

**Summary:**

In this paper, the authors focus on a more realistic setting for temporal causal representation learning. Instead of assuming that the underlying mechanisms only switch between a few discrete domains, they consider the case where they change continuously over time.
To deal with this, the authors propose a two-stage method called TRACE.
First, it learns a set of basic mechanisms, and then it recovers the smooth changes in how these mechanisms are mixed over time. The paper also gives theoretical results on recovery and shows promising experimental results.

**Compliance With Llm Reviewing Policy:**

Affirmed.

**Final Justification:**

As shown in the rebuttal, the authors address my concern and I maintain my score.

**Key Questions For Authors:**

How robust is TRACE when its main assumptions are violated, e.g. a). when pure-domain data are unavailable; b). the number of atomic mechanisms is misspecified?

**Limitations:**

Yes.

**Strengths And Weaknesses:**

Strengths:
1. The problem setting is meaningful, and the idea is novel. Continuous changes in mechanisms seem more realistic.

2. The method is clearly designed, and the authors provide theoretical insights.

3. The experiments are comprehensive and generally support the main claim of the paper.

Weakness:

1. The assumptions are still fairly strong, such as knowing the number of mechanisms and requiring rank conditions.

2. The real-world experiments use proxy labels instead of true ground-truth mechanisms.

3. Some of the baseline comparisons are not fully fair, since they were not designed to recover continuous trajectories.

---

> ### Author Rebuttal · Authors · 2026-03-30
>
> We thank Reviewer i2hJ for the thoughtful evaluation. We address each concern with new experiments.
>
> **W1/Q(a): When pure-domain data are unavailable.**
>
> We test two types of domain impurity:
>
> (a) *Label noise.* We randomly flip 20% of training samples' domain labels to a uniformly chosen incorrect domain. MCC remains >0.96 and Weight Corr (trajectory-level Pearson correlation) remains >0.96. Label noise affects only expert routing during training; the shared encoder's reconstruction objective is label-agnostic, and individual experts tolerate moderate contamination from mis-routed samples.
>
> (b) *Mechanism impurity* ($K_{\text{active}}=3, K_{\text{total}}=5$), a stronger test at the data generating process level. We contaminate each domain's transition matrix: $W_k = W_{\text{base}} + (1-\varepsilon)\,\delta W_k + \varepsilon \cdot \text{mean}(\delta W_{j \neq k})$ where all entries of $\delta W$ are nonzero:
>
> | $\varepsilon$ |     MCC     | Weight Corr | $\sigma_{\min}$ |
> | :-----------: | :---------: | :---------: | :-------------: |
> |      0%       | 0.984±0.016 | 0.996±0.001 |      0.168      |
> |      20%      | 0.975±0.002 | 0.981±0.002 |      0.126      |
> |      50%      | 0.943±0.032 | 0.936±0.033 |      0.049      |
>
> Weight Corr **= 0.981 for $\varepsilon = 20\%$** and **0.936 for $\varepsilon = 50\%$**. Degradation is gradual and consistent with Theorem 4.2: as $\sigma_{\min}$ (the minimum singular value of the basis matrix $\hat{B}$, measuring domain separability) decreases monotonically from 0.168 to 0.049, the error bound $O(1/\sigma_{\min})$ predicts proportionally worse recovery. Under this contamination model, each domain's specific signal vanishes at $\varepsilon = (K_{\text{total}}-1)/K_{\text{total}}$; with $K_{\text{total}}=5$, domains remain distinguishable until $\varepsilon \approx 80\%$, so $\varepsilon = 50\%$ is well below the critical threshold.
>
> In summary, TRACE is robust to both label-level and mechanism-level impurity. $\sigma_{\min}$ serves as a practical pre-deployment diagnostic for domain separability.
>
> **W1/Q(b): When $K$ is misspecified.**
>
> We test training-time misspecification with $K_{\text{total}}=5$:
>
> (i) *Over-specification* ($K_{\text{train}}=7$, i.e. 2 extra domains): We test duplicate domains (copying existing mechanisms) and mixed domains (using averaged $W$ matrices), using all 7 domains as basis.
>
> When the test trajectory's active domains do not include duplicated experts, Weight Corr = **0.966**, close to the $K_{\text{train}}=5$ baseline (0.971). The slight degradation reflects additional near-duplicate columns in $\hat{B}$. When active domains include duplicated experts, $\alpha$ recovery degrades (Weight Corr = 0.553) because the solver cannot distinguish near-identical basis vectors. However, the recovered transition dynamics $W(t)$ remain accurate, since duplicate experts share identical $W^{(k)}$, so any $\alpha$ split between them yields the same $W(t) = \sum_k \alpha_k(t) W^{(k)}$. Practitioners can detect and merge duplicates via $\sigma_{\min}$-based analysis or post-hoc clustering.
>
> (ii) *Under-specification* ($K_{\text{train}}=3$, 2 missing domains): Recovery for trajectories involving only seen domains remains strong (Weight Corr = **0.949**). For trajectories involving unseen domains, neither $\alpha$ nor $W(t)$ can be recovered (Weight Corr = 0.244), as their dynamics were never learned. This is an inherent information limitation.
>
> In summary, over-specification causes only minor degradation (0.966 vs 0.971), and under-specification degrades gracefully for seen domains (0.949). We recommend mild over-specification combined with $\sigma_{\min}$-based duplicate detection.
>
> **W2: Proxy labels in real-world experiments.**
> Ground-truth $\alpha(t)$ is unavailable for real-world data, so we use physically motivated proxies. For the vehicle dataset, the proxy is the ratio of vertical to total velocity ($|v_y|/(|v_x|+|v_y|)$), which is approximately linearly related to $\alpha_{\text{turn}}$ within its operating range. For the gait dataset, hip joint speed serves as the proxy, increasing approximately linearly with the walk-to-run mixing coefficient. Pearson correlation is therefore a valid evaluation metric.
>
> **W3: Baseline comparisons.**
> We agree that the baselines were designed for discrete switching. To our knowledge, no existing method can jointly recover latent variables and continuous mechanism mixing trajectories, which is precisely the novel problem TRACE addresses. We include the most relevant discrete-switching baselines to highlight the structural gap: NCTRL's oscillation between discrete states (Figure 5b in the main paper) illustrates that discrete methods are structurally ill-suited to capture continuous transitions, which motivates our formulation.
>
> All new experiments and a strengthened limitations discussion (covering the pure-domain requirement, K-misspecification trade-offs, and scale constraints) will be included in the revision.

---

> > ### Author Rebuttal · Reviewer_i2hJ · 2026-04-02
> >
> > Thanks for the rebuttal. My concern is solved. I maintain my score.

---

> > > ### Author Response · Authors · 2026-04-03
> > >
> > > We sincerely thank Reviewer i2hJ for the constructive feedback and for confirming that the concerns have been addressed. All new experiments on domain impurity and K-misspecification, along with the strengthened limitations discussion, will be incorporated in the revised manuscript.

---

### Decision · Program_Chairs · 2026-04-30

**Decision:**

Accept (regular)

**Comment:**

This paper introduces TRACE, a framework for temporal causal representation learning where causal mechanisms evolve continuously rather than switching discretely. Mechanisms are modeled as convex combinations of K atomic mechanisms with time-varying mixing coefficients over the simplex, learned via a two-stage approach: atomic mechanism discovery from pure-domain data, then continuous trajectory recovery via least-squares projection with temporal smoothing. The paper provides identifiability results with finite-sample error bounds and evaluates on synthetic data, CartPole, UAV dynamics, and gait transitions.

The problem formulation is novel and addresses a genuine gap, as existing methods assume discrete switching while many real systems exhibit continuous transitions. Theoretical foundations are strong, covering both latent-variable and mixing-trajectory identifiability with explicit error characterization. All four reviewers maintained or raised scores after the rebuttal. Key concerns were convincingly addressed: the geometric capacity constraint (K ≤ d+1) was shown empirically to be sufficient but not necessary, and contamination experiments demonstrated graceful degradation under domain impurity (Weight Corr = 0.981 at 20%, 0.936 at 50%).

This paper makes a clear contribution by formalizing the evolution of continuous mechanisms with strong theoretical guarantees and extensive empirical validation. The remaining limitations (pure-domain requirement, geometric constraint, proxy-based evaluation) are acknowledged and do not undermine the core contribution.